# Disentangling plant- and environment-mediated drivers of active rhizosphere bacterial community dynamics during short-term drought

Sreejata Bandopadhyay [1,2,3], Xingxing Li [3,4], Alan W. Bowsher[1,2], Robert L. Last [3,4,5] & Ashley Shade [6] ✉

Mitigating the effects of climate stress on crops is important for global food security. The microbiome associated with plant roots, the rhizobiome, can harbor beneficial microbes that alleviate stress, but the factors influencing their recruitment are unclear. We conducted a greenhouse experiment using field soil with a legacy of growing switchgrass and common bean to investigate the impact of short-term drought severity on the recruitment of active bacterial rhizobiome members. We applied 16S rRNA and 16S rRNA gene sequencing for both crops and metabolite profiling for switchgrass. We included planted and unplanted conditions to distinguish environment- versus plant-mediated rhizobiome drivers. Differences in community structure were observed between crops and between drought and watered and planted and unplanted treatments within crops. Despite crop-specific communities, drought rhizobiome dynamics were similar across the two crops. The presence of a plant more strongly explained the rhizobiome variation in bean (17%) than in switchgrass (3%), with a small effect of plant mediation during drought observed only for the bean rhizobiome. The switchgrass rhizobiome was stable despite changes in rhizosphere metabolite profiles between planted and unplanted treatments. We conclude that rhizobiome responses to short-term drought are crop-specific, with possible decoupling of plant exudation from rhizobiome responses.

The idea that beneficial microbes can be managed to improve crop performance is building momentum with new research directions[1–4]. It is well known that plant-associated microorganisms can promote plant performance when conditions are unfavorable[5]. For example, microbes can help plants acquire limiting nutrients, such as nitrogen and phosphorus[6]. In turn, plants can shape their microbiome by secreting metabolites, including sugars in their root exudates to support microbiome recruitment and maintenance[7]. Furthermore, different plant genotypes can have distinct metabolite compositions[8], with potential consequences for microbiome recruitment[9]. In the

[1]Department of Microbiology, Genetics, and Immunology, Michigan State University, East Lansing, MI, USA. [2]Plant Resilience Institute, Michigan State University, East Lansing, MI, USA. [3]U.S. Department of Energy Great Lakes Bioenergy Research Center, Michigan State University, East Lansing, MI, USA. [4]Department of Biochemistry and Molecular Biology, Michigan State University, East Lansing, MI, USA. [5]Department of Plant Biology, Michigan State University, East Lansing, MI, USA. [6]Universite Claude Bernard Lyon 1, CNRS, INRAE, VetAgro Sup, Laboratoire d'Ecologie Microbienne LEM, CNRS UMR5557, INRAE UMR1418, Villeurbanne F-69100, France. ✉e-mail: ashley.shade@cnrs.fr

bioenergy feedstock switchgrass (*Panicum virgatum*), there are different metabolite profiles for upland and lowland ecotypes[8], with in vitro studies showing that differential metabolite accumulation across ecotypes can contribute to different microbiome compositions[9].

The increased frequency and intensity of droughts have become major challenges for crop production globally[10,11]. During drought, soil loses its moisture content, which can be exacerbated by erratic rainfall and temperature fluctuations[12]. There also are changes in other soil and plant properties, such as root exudation[13], with concurrent changes in the composition and function of the local soil and root-associated microbiome. Microorganisms can also play direct or indirect roles in plant drought tolerance. For example, some plant growth-promoting bacteria exude exopolysaccharides, which can retain the soil moisture content at the beginning of drought[14]. They can also produce antioxidant enzymes during drought that can support plant response to reactive oxygen species resulting from drought-related stress[15]. However, beneficial and detrimental strains that affect plant growth under drought conditions have been identified (for example, in sorghum[16]). Drought can increase the relative abundance of phyla such as Firmicutes, Chloroflexi, and Actinobacteria, while phyla such as Bacteroidetes and Planctomycetes are relatively depleted[13]. Finally, techniques such as host-mediated microbiome engineering has also been employed in crops, such as wheat, to select beneficial microbial communities that promote plant tolerance to drought stress[17].

Drought responses of the rhizosphere and endosphere microbiomes have been extensively studied in crops such as rice[18,19] and sorghum[20]. These studies demonstrated that drought can cause taxonomic enrichment of bacteria that may help plants recover during stress while also affecting the temporal assembly of the root microbiome. A recent study used random forest models to show that different drought regimes altered microbiome succession as compared to the watered controls[18]. In this study, the degree of difference in the microbiome was directly proportional to the duration of drought, and the delay in microbiome succession persisted even after watering resumed, with enduring changes in the root microbiome observed as many as 62–84 days after the drought ended[18].

Because of the typically low microbial biomass associated with dry soil conditions, it remains technically challenging to assess the active microbial members of the rhizosphere during drought, and few studies have done so[20]. As the dormant bacterial pool in soils and rhizospheres is substantial[21–23], separating the responses of active from dormant members may be insightful for targeting the most responsive populations for plant benefit. Additionally, different plants have different inherent tolerances to dry conditions. Thus, comparing the rhizosphere microbiomes across differently drought-sensitive crops may also provide insights into the community members that can benefit sensitive plants. Furthermore, while changes in plant root exudation during drought can shape the rhizosphere microbiome response[24–26], it is unclear how much of that response is mediated by the host or attributable to the environmental conditions of drought, such as low water availability, that impact the microbes directly. Drought conditions include low moisture availability, changes in physical soil structure, and connectivity via soil pores, all of which can directly impact microbes. Understanding whether responses are host- or environment-mediated could inform the separate targets for microbiome modification via host and soil management. Changes in rhizosphere exudates during drought stress could also serve as signals to reactivate and recruit members from the dormant microbial pool to the rhizosphere, though this has not been thoroughly considered.

In rhizosphere soils or similar environments containing ample amounts of energy-rich substrates, it is thought that much of the microbial community is active rather than dormant[27,28]. Evidence from [13]C-PLFA (phospholipid-derived fatty acids) shows that Gram-positive bacteria assimilated [13]C more actively in the rhizosphere than in bulk soil[29]. Studies have also shown that Gram-negative bacteria in the rhizosphere actively assimilated root-derived carbon more successfully than Gram-positive bacteria[30–33]. For example, activity staining of bacterial cells from rhizosphere soil revealed that as much as 55% of them were active, with further evidence suggesting that rhizosphere soils have approximately 20% more active cells than bulk soils[28,34]. However, the active rhizosphere microbiome pool can change during stress events, with members shifting to dormancy in response to the changing environment or climate, for instance, low moisture availability. The contribution of the resuscitated or active community becomes more prominent during and in the immediate aftermath of stress, when environmental conditions fluctuate, and prevalent members shift in competitiveness. Thus, it is critical to understand active microbial community dynamics over short-term stress exposure.

Several knowledge gaps exist about microbiome activation and recruitment during drought. First, the factors affecting the recruitment of the active rhizosphere microbiome during drought are unclear; specifically, the roles of the droughted environmental conditions and plant as co-occurring but distinct drivers in that assembly. While there are several studies that investigate plant microbiome outcomes after long-term drought exposure[35,36], there is limited knowledge about which microbiome members are quickly responsive to the immediate effects of the drought; these responsive members include those that are both positively and negatively selected by the drought, the plant, or their interaction, and therefore play a key role in the re-initiation of community assembly. Finally, how these microbiome dynamics differ across plant families with different drought tolerances and to what extent the microbiome response is attributable to the host or environment needs to be understood.

To address these knowledge gaps, we conducted a greenhouse experiment using the annual legume common bean (*Phaseolus vulgaris, var.* Red Hawk) and the perennial grass switchgrass (*Panicum virgatum, var.* Cave-in-Rock). These crops substantially differ in physiology and root architecture, as bean has a taproot system, and switchgrass has a dense rhizome that can extend several meters belowground. Furthermore, switchgrass genotypes are relatively more consistently drought-resistant, whereas bean genotypes have more variation in their drought tolerance[37–40]. The objectives of this study were to understand the active bacterial rhizosphere microbiome immediate assembly and short-term recruitment over a gradient of drought severity. For both plants, we compared planted to unplanted conditions using agricultural field soils previously planted with one of the two crops (and thus contained their rhizosphere microbiome as a legacy) and partitioned the influences of plant-mediated versus environment-mediated rhizobiome responses to drought. We henceforth use the term "rhizobiome" throughout, even when comparing planted and unplanted treatments, to reiterate that the field soils used in the experiment were selected because they were root-influenced by the same crop as the planted condition. We addressed two hypotheses. First, we hypothesized that active rhizobiomes respond to drought and change progressively with drought severity. Second, we hypothesized that the host plant mediates the responses of the active rhizobiome to drought via compositional changes in root metabolites.

## Results

### Overview

We conducted a short-term drought experiment over 6 days and assessed the active rhizobiome dynamics of bean and switchgrass in the greenhouse. Rhizosphere soils were collected from fields recently planted with each crop, and treatments included planted/unplanted and watered/drought. We collected pre-drought and post-drought samples over time and with increased drought severity (Fig. 1).

### Efficacy of the short-term drought treatment

The watered samples in both bean and switchgrass had relatively high and stable gravimetric soil moisture content over the different time

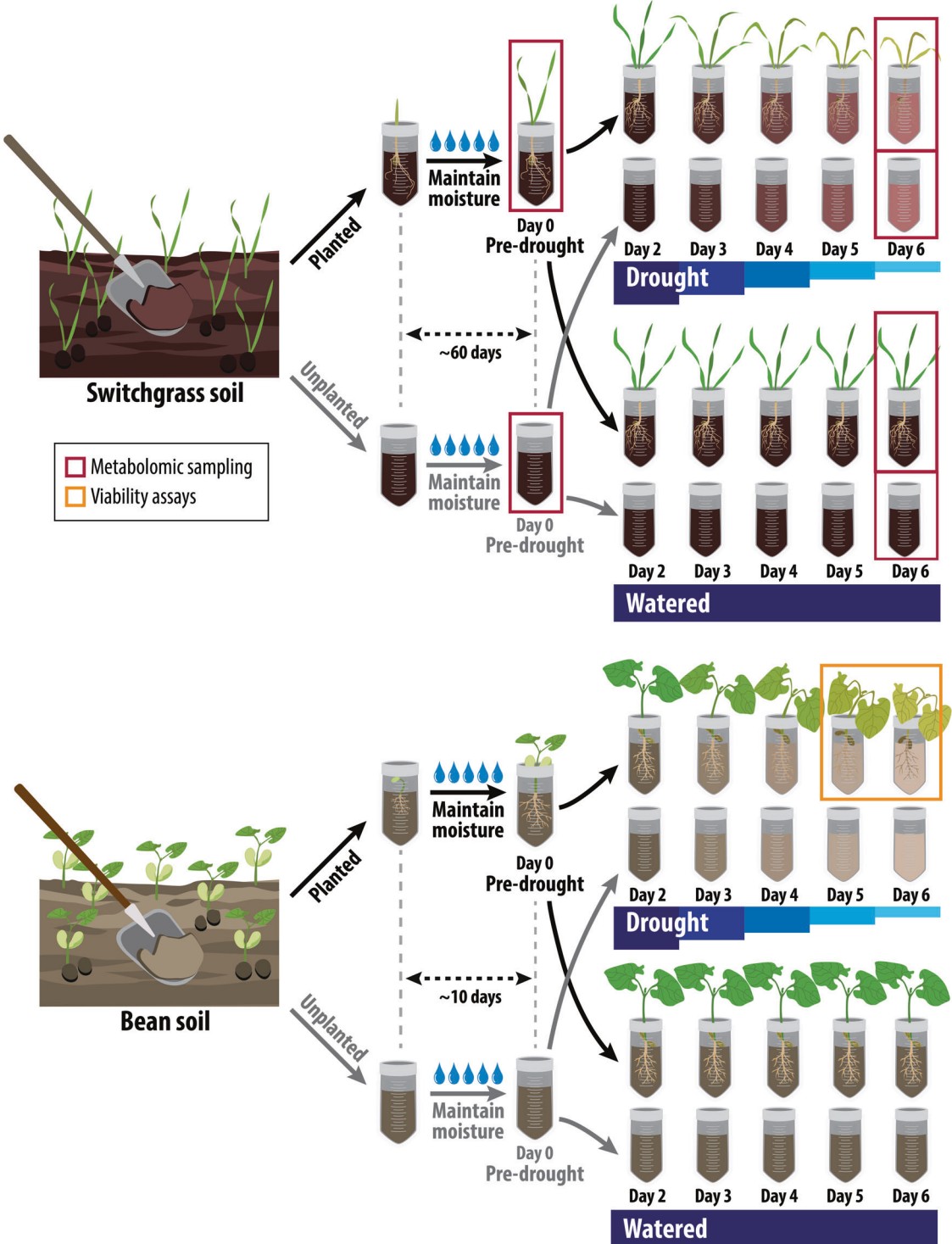

**Fig. 1 | The experimental design used in the greenhouse study.** Day 2 to day 6 correspond to increased drought severity. Prior to the initiation of treatments, switchgrass plants were close to the flowering stage after ~60 days, but no flowers were observed. Bean plants were in their vegetative stage after ~10 days, with the first trifoliate leaf unfolded in all plants. We conducted metabolomics for switchgrass because we found the switchgrass rhizobiome was relatively stable to drought treatment and wanted to know if there was evidence of a plant drought response via metabolite output. We conducted the viability assays for bean because our pilot experiments indicated that the bean plants may lose turgor by days 5 and 6 and we wanted to ensure investigation of rhizobiome responses to stressed, live plants (rather than necromass). Source data are provided as a Source Data file.

points for both planted and unplanted conditions (Fig. 2A). In bean, the drought samples had a decrease in moisture content over time for both planted and unplanted soils, as expected, while in switchgrass, drought samples had comparable moisture content over time for both planted and unplanted soils. The final sample (day 6) for switchgrass had the lowest mean soil moisture in planted soils and the highest mean moisture in unplanted soils. In both bean and switchgrass, the unplanted drought treatments retained higher soil moisture than the planted ones. There were interaction effects of drought, sampling day, and plant presence on bean soil moisture (Three-way ANOVA type III

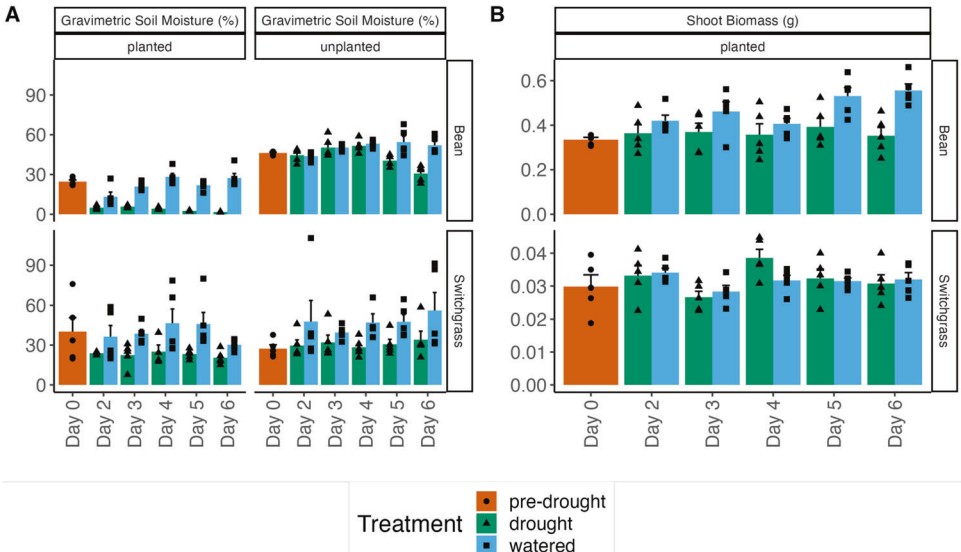

**Fig. 2 | Gravimetric soil moisture and shoot biomass of bean and switchgrass before and after drought treatment. A** Gravimetric soil moisture of bean (sample size [$n$]=110) and switchgrass (sample size [$n$]=110) planted and unplanted soils at different time points for drought and watered treatments. **B** Shoot biomass of bean (sample size ($n$)=55) and switchgrass (sample size ($n$) = 55) at different sampling days for drought and watered treatments. Bars denote the mean of five replicate measurements, error bars denote standard error of the mean. Points overlaid on top of the graphs denote individual data points for each treatment-time combination. Source data are provided as a Source Data file.

test, $F = 2.84$, $P = 0.03$, Supplementary Table 1), suggesting the effects of each of these factors depended on the levels of the other factors. In switchgrass, there were main effects of drought on soil moisture (three-way ANOVA Type III test, $F = 28.43$, $P < 0.001$) and of plant presence on soil moisture ($F = 6.66$, $P = 0.01$).

In bean plants, the shoot biomass was always higher in the watered samples as compared to drought (two-way ANOVA Type III test, $F = 21.18$, $P < 0.001$), and the watered plants increased in biomass over time (Fig. 2B). There were no differences in switchgrass biomass between watered and drought samples over time (two-way ANOVA, $F = 0.30$, $P = 0.59$). Switchgrass shoot biomass fluctuated mildly over time but had no obvious trend (two-way ANOVA Type III test, $F = 3.43$, $P = 0.02$, Supplementary Table 1).

## Metabolome responses of drought for switchgrass

Previous work has shown that switchgrass alters its metabolism during drought[36]. Given the absence of switchgrass shoot biomass response to the drought, we performed untargeted LC-MS-based metabolomics of switchgrass soils to assess any drought-induced changes in plant metabolites that could impact its microbiome. In addition, we also profiled the global changes that occurred in the metabolomes of switchgrass shoots and roots caused by the drought treatment. The metabolite profiling revealed 3532 distinct metabolite features whose maximum abundance among the biological samples was ≥ 500 counts (Supplementary Data 1–3).

A comparative analysis revealed 1051 root-, 538 shoot- and 231 soil-specific features (Supplementary Fig. 1A). The root and shoot shared 1551 features, the root and soil shared 883 features, and the leaf and soil shared 750 features, while 736 features were shared by all three groups (Supplementary Fig. 1A). The planted rhizosphere soils differed from the unplanted ones and the planted soil collected on day 0 (PERMANOVA, planted versus unplanted, $F = 11.26$, R$^2 = 0.27$, $P = 0.001$, Supplementary Fig. 1B). There also were differences between day 0 and day 6 soils, indicating an overall effect of time (PERMANOVA $F = 3.55$, R$^2 = 0.1$, $P = 0.02$).

While the scatter plot suggested divergence between the drought and watered metabolomes for planted soils (Supplementary Fig. 1C), these overarching trends were not statistically supported (PERMANOVA $F = 2.2$, $R^2 = 0.15$, $P = 0.07$). However, several metabolite features strongly differentiated the treatments. Saponins most differentiated the switchgrass metabolite soil profiles across the planted and drought samples (the metabolite annotation confidence levels can be found in Supplementary Data 4). Among the discriminating features, *10.11_1000.5271n* was the most explanatory (highest variable of importance, VIP, Fig. 3). However, the low abundance of this feature detected in the rhizosphere soil samples caused a low-intensity and therefore not clear MS/MS mass spectra (Supplementary Fig. 2A). We speculate that it is a previously identified steroidal saponin with a single sugar-chain from switchgrass root tissue extracts[8] (Supplementary Fig. 2B). Some other most important features were also annotated as specialized metabolites, including the root-accumulating diterpenoids with elevated concentrations in drought-treated planted soils (Supplementary Fig. 1D, Supplementary Data 4). This finding is consistent with the studies investigating longer drought exposure[8,36]. The higher abundances of these specialized metabolites in the drought samples from the planted soils suggest that switchgrass could release them into the rhizosphere soil when stressed by drought. Additional metabolite results are provided in the Supplementary Results section.

## No detection of relative changes in rhizobiome size over time

To proxy relative changes in microbiome size or microbial biomass, we considered the recovered DNA concentration from the rhizosphere soil before dilution and normalization for sequencing[41,42]. Linear models suggested that there were no supported trends in DNA concentrations over time for any of the treatments (Supplementary Fig. 3, all $P > 0.05$).

## Rhizobiome sequencing summary: total and active bacteria communities

For the total community (DNA) data, total non-chimeric merged reads for both bean and switchgrass were 20,526,147. 19,938,535 reads remained after filtering out mitochondria, chloroplast, and contaminant OTUs, out of which 10,204,741 and 9,733,794 reads were attributed to switchgrass and bean, respectively. For the active community (cDNA) data, total non-chimeric merged reads for both bean and switchgrass were 22,018,663, of which 21,796,243 reads remained

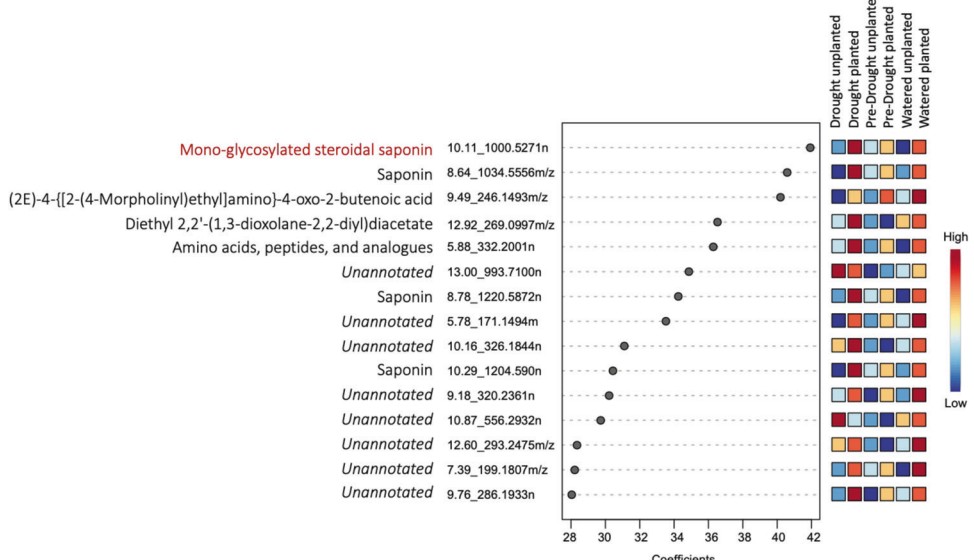

**Fig. 3 | VIP (variable of importance) coefficient score plot from a PLS-DA (partial least-squares discriminant analysis) model showing the top 15 variables (features) of importance to differentiate the different switchgrass soil types.** Feature *10.11_1000.5271n*, a previously identified switchgrass saponin,

had the highest coefficient and is thus the most important detected variable during the drought treatment. At the end of each feature name, 'm/z' stands for mass-to-charge ratio and 'n' stands for neutral mass. Source data are provided as a Source Data file.

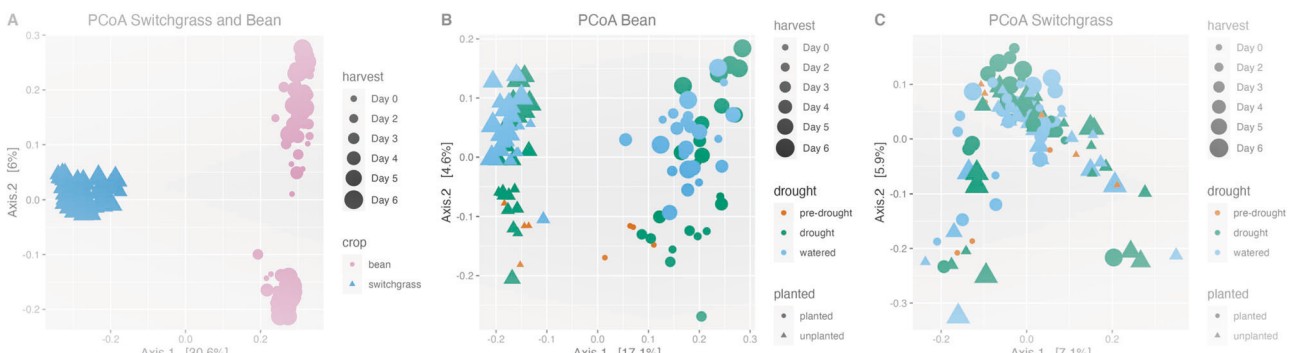

**Fig. 4 | Beta diversity showing differences in active bacterial rhizobiome communities across bean and switchgrass, and treatments within each crop.** **A** Principal Coordinate Analysis (PCoA) comparing bean and switchgrass active community composition (sample size ($n$)=212). PCoA of **B** bean (sample size ($n$)=105)

and **C** switchgrass (sample size ($n$)=107) comparing active communities of planted and unplanted soil samples, as well as comparisons across drought treatments: pre-drought, drought, and watered. Samples collected across all sampling days are shown. Source data are provided as a Source Data file.

after filtering out mitochondria, chloroplast, and contaminant OTUs, out of which 11,799,699 and 9,996,544 reads were attributed to switchgrass and bean, respectively. The total number of taxa (DNA) and active taxa (based on DNA/RNA ratios) were 21,407 and 8732, for bean samples and 17,331 and 7539, respectively, for switchgrass samples.

The microbiome data presented, henceforth, all pertain to the DNA counts of the active community members (16SrRNA:rRNA gene >= 1). The percent active taxa ranged from 40 to 60% of the total observed taxa in a sample. This establishes that an estimated half of the taxa detected in the DNA dataset were not observed or did not meet our activity criterion in the RNA dataset.

## Bean and switchgrass had distinct active rhizobiomes
The literature shows that different plants harbor different microbiomes[43–45]. Here, we also found different total (PER-MANOVA Psuedo-F = 163.02, $R^2$ = 0.44, $P$ = 0.001) and active rhizo-biome structures for bean and switchgrass (Supplementary Fig. 4, Fig. 4A, active community PERMANOVA Pseudo-F = 91.21, $P$ = 0.001, $R^2$ = 0.30), with 3883 active OTUs shared between the crops. However,

the overall beta dispersion was comparable across bean and switch-grass (PERMDISP $F$ = 2.95, $P$ = 0.09).

## Addressing hypothesis 1: Active rhizobiomes respond to drought and change progressively with drought severity
Effects of drought, sampling day (colinear with drought severity for the drought treatment), and drought: sampling day interactions were significant and of similar explanatory value for both bean and switch-grass (Fig. 4B, C, PERMANOVA all $P$ < 0.05, see Supplementary Table 2). We further investigated the effect of the drought treatments (levels: drought, pre-drought, and watered) using post hoc tests. In switch-grass, there were differences between pre-drought and watered treatments (PERMANOVA Pseudo-F = 1.36, $R^2$ = 0.02, $P$ = 0.03) and between the watered and drought condition (Pseudo-F = 1.55, $R^2$ = 0.02, $P$ = 0.004). In bean, there were differences between the pre-drought and drought condition (Pseudo-F = 1.79, $R^2$ = 0.03, $P$ = 0.02), between the pre-drought and watered condition (Pseudo-F = 2.27, $R^2$ = 0.04, $P$ = 0.007), and between the drought and watered condition (Pseudo-$F$ = 2.1, $R^2$ = 0.02, $P$ = 0.006).

In bean samples, planted rhizobiomes had lower richness (number of active OTUs) than unplanted ones (Three-way ANOVA, $F = 60.12$, $P < 0.001$, Supplementary Table 3). For switchgrass, there were no differences in richness between the levels of different factors tested in the experiment, except for a weak interaction observed between planted and drought treatments (Three-way ANOVA $F = 4.30$, $P < 0.05$, Supplementary Table 3).

We hypothesized that rhizobiomes in the drought samples would change faster and to a greater extent than the watered ones. We also hypothesized that the presence of the plant might suppress or stabilize these dynamics due to host-microbiome feedback, resulting in reduced and slower change in planted than unplanted soils. To test this, we assessed linear models of the active rhizobiome structure (beta diversity) over time and as compared to the pre-drought condition (Supplementary Fig. 5). In these models, the slopes represent the rates of change, and the intercepts represent their magnitudes. However, the linear models generally were not supported for changes in beta diversity, suggesting only incremental beta diversity changes given the short-term drought. The exception was detecting significant linear regression for changes in beta diversity in the switchgrass watered and planted rhizobiome ($R^2 = 0.23$, $P = 0.02$).

While there were no detected differences in beta-dispersion (community variance) among switchgrass experimental factors, for bean rhizobiomes, there were differences detected among the drought treatments, with post-hoc tests revealing differences in dispersion between the watered and drought rhizobiomes (mean dispersion(watered) = 0.41, mean dispersion (drought) =0.44, PERMDISP Pseudo-F = 13.56, d.f.=1, $P = 0.001$) and between the pre-drought and drought rhizobiomes (mean dispersion (pre-drought) = 0.40, mean dispersion (drought) = 0.44, PERMDISP Psuedo-F = 8,15, d.f.= 1, $P = 0.01$), but not between the pre-drought and watered (PERMDISP $P = 0.47$).

There were statistically supported differences in community structure with drought and time/drought severity for both plants. Still, it is notable that there were no apparent differences in the temporal dynamics (rate and magnitude of change) across these treatments. In addition, average distance to the median in community structure (dispersion) was modestly higher in droughted bean rhizobiomes than in pre-drought and watered ones. Thus, hypothesis 1 was supported by the data.

**Addressing hypothesis 2: The plant mediates the responses of the active rhizobiome members to drought**

The most striking difference between the rhizobiomes of the two crops was whether a plant was present. Though the presence of a plant was a significant explanatory factor for both crops' rhizobiomes, it had the highest explanatory value of all the factors tested for bean (17% explanatory value, Fig. 4B) and relatively low explanatory for switchgrass (3% explanatory value, Fig. 4C, Supplementary Table 2). Furthermore, the interaction between plant presence and drought was only significant in bean rhizobiomes (PERMANOVA Pseudo-F = 1.84, $P = 0.03$) and not in switchgrass (PERMANOVA Pseudo-F = 1.13, $P = 0.18$, Supplementary Table 2). We conclude that whether the plant mediates rhizobiome responses to short-term drought depends on the plant. Thus, hypothesis 2 was partially supported and conditional on the plant species investigated. There were plant-mediated responses to drought for bean rhizobiomes but not for switchgrass.

Motivated by the plant-mediated effects of drought for bean, we next investigated the activated taxa that were uniquely associated with the planted drought condition in bean rhizobiomes. We identified 87 such taxa using indicator analysis (Fig. 5A). We reasoned that the responses of these activated taxa are likely to be mediated, directly or indirectly, by the bean plant during drought.

Focusing on the most abundant 50 indicators of the planted drought condition, we examined their activity dynamics over drought severity (Fig. 5B). Several active OTUs increased over time and with drought severity (enriched in Days 5-6, within clades 1 and 3), including members belonging to the classes Actinobacteria, Verrucomicrobiae, Acidobacteriae, Acidimicrobiia, Thermoleophilia, and Alphaproteobacteria. Several OTUs were detected consistently over time despite increasing drought severity (clade 4), including some belonging to Alphaproteobacteria, Planctomycetes, Gammaproteobacteria, and Actinobacteria (Fig. 5B). These planted drought indicator taxa that were either enriched over time or stable are targets for follow-up research to understand their exact fitness advantages and potential benefits to the host in the drought environment.

Notably, while the indicator species analysis revealed several OTUs strongly associated with each drought and planting level for bean rhizobiomes, no indicators were discovered for switchgrass, further supporting the previous finding of no interaction of planting and drought for switchgrass.

In the drought samples planted with bean, we noted that many stable or enriched OTUs belonged within few classes. Thus, we next compared the overall differences and changes in the relative abundances of these classes between the treatments for bean rhizobiomes (Fig. 6). One motivation for this analysis was to understand what classes the plant retains during drought by comparing the planted and unplanted drought conditions. Only taxa associated with Blastocatellia were absent in the unplanted drought versus the planted drought rhizobiomes. However, Gammaproteobacteria (Kruskal-Wallis chi-squared = 23.96, df = 1, $P < 0.001$), Actinobacteria (Kruskal-Wallis chi-squared = 5.89, df = 1, $P = 0.02$) and Alphaproteobacteria (Kruskal-Wallis chi-squared = 26.75, df = 1, $P < 0.001$) were significantly higher in relative abundance in the planted compared to the unplanted treatments in bean under drought condition. Among these, Gammaproteobacteria (Kruskal-Wallis chi-squared = 31.59, df = 1, $P < 0.001$) and Alphaproteobacteria (Kruskal-Wallis chi-squared = 16.65, df = 1, $P < 0.001$) were significantly increased in planted compared to unplanted treatments in watered bean samples. At the same time, Actinobacteria was not statistically different between planted and unplanted treatments.

Another motivation was to identify potentially "lost" taxa during a drought that are normally associated with the bean. We explored drought-sensitive taxa by comparing the planted drought to the planted watered samples, reasoning that if any of these taxa are beneficial for the plant, they could be targets to prioritize for recovery after a drought. However, there were no notable absences or decreases of the major active classes in the planted soils across watered and drought conditions. Rather, the most notable comparative difference among treatment groups was between the planted and unplanted conditions, as several classes were only associated with the unplanted soils, including vadinHa49, Subgroup 5, Pla4_lineage, Oligoflexia, FCPU426, and bacteriap25. These classes are potentially selected against or non-competitive in the presence of the plant.

## Discussion

Soil microbes can be important in plant drought tolerance[46]. Improving our understanding of root-associated soil microbial responses during drought stress can help improve a plant's resilience through targeted microbiome manipulation. This study used microbiome and metabolomics data to understand microbial community shifts during short-term drought in two plant species – common bean and switchgrass. We focused on the subset of the bacterial community that was likely to be active (RNA-based assessment) during drought by comparing unplanted and planted rhizosphere soils to watered soils. These comparisons were critical to tease apart plant-mediated responses to drought, which otherwise can be challenging to explore as the stress of drought independently will alter soil microbial communities and direct many taxa to dormancy.

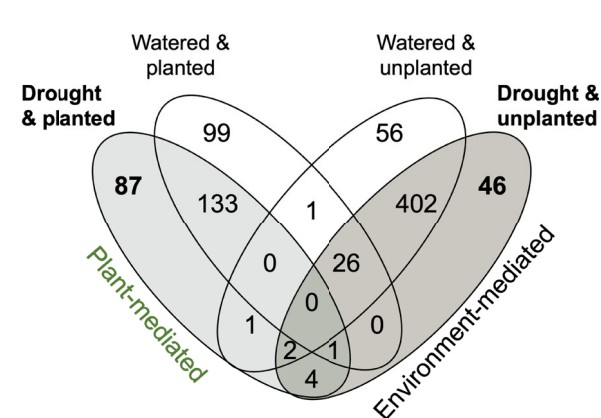

## A  Activated Taxa in Bean

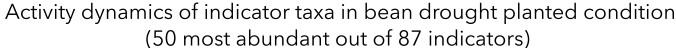

Activity dynamics of indicator taxa in bean drought planted condition
(50 most abundant out of 87 indicators)

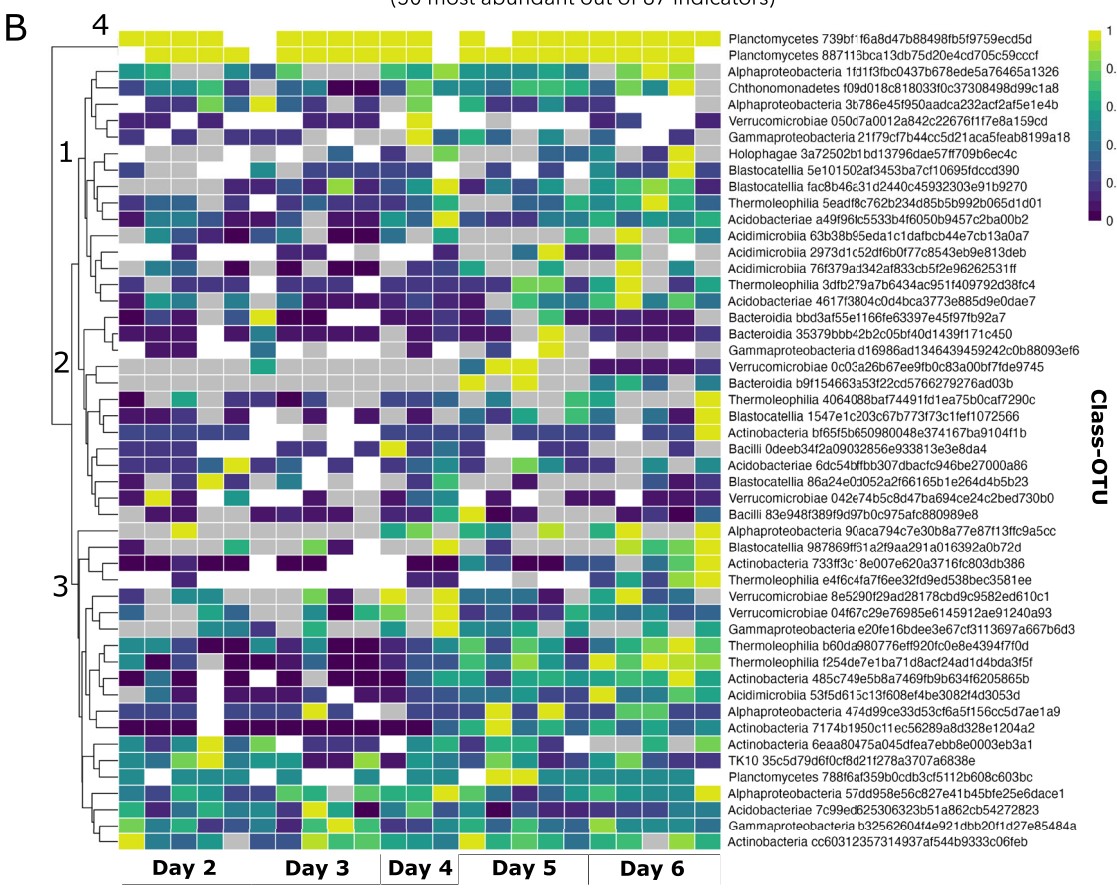

**Fig. 5 | Active indicator taxa associated with the bean crop for each treatment of the drought experiment along with the activity dynamics of the most abundant indicator taxa observed in bean planted drought treatment. A** Venn diagram showing the indicator species associated with each drought/planted condition in common bean (sample size (*n*)=96). **B** Heatmap showing the relative abundance dynamics of the top 50 most abundant and active OTUs from the 87 indicators associated with the bean planted drought condition. Inactive taxa are coded as "0" (zero), corresponding to the light gray cells. More abundant OTUs are more yellow, and less abundant are bluer. Taxa not detected in either DNA or RNA dataset is denoted as NA and colored as white in the heatmap. We standardized counts within an OTU relative to the maximum observed abundance value detected for that OTU across samples. Phantom taxa detected in >5% of samples were included. The legends on the right indicate classification at the Class level and the OTUID (sample size (*n*)=23). Source data are provided as a Source Data file.

Soil gravimetric moisture content confirmed that the drought treatment was effective for both crops in that droughted soil had less moisture than watered treatments, per crop. Given that bean and switchgrass have very different water needs, which was one of the motivations for their use in this experiment, we intentionally provided them different amounts of water to achieve a relatively equivalent stress to each crop, as informed by a pilot experiment we executed to optimize growth parameters for each crop. The soil moisture

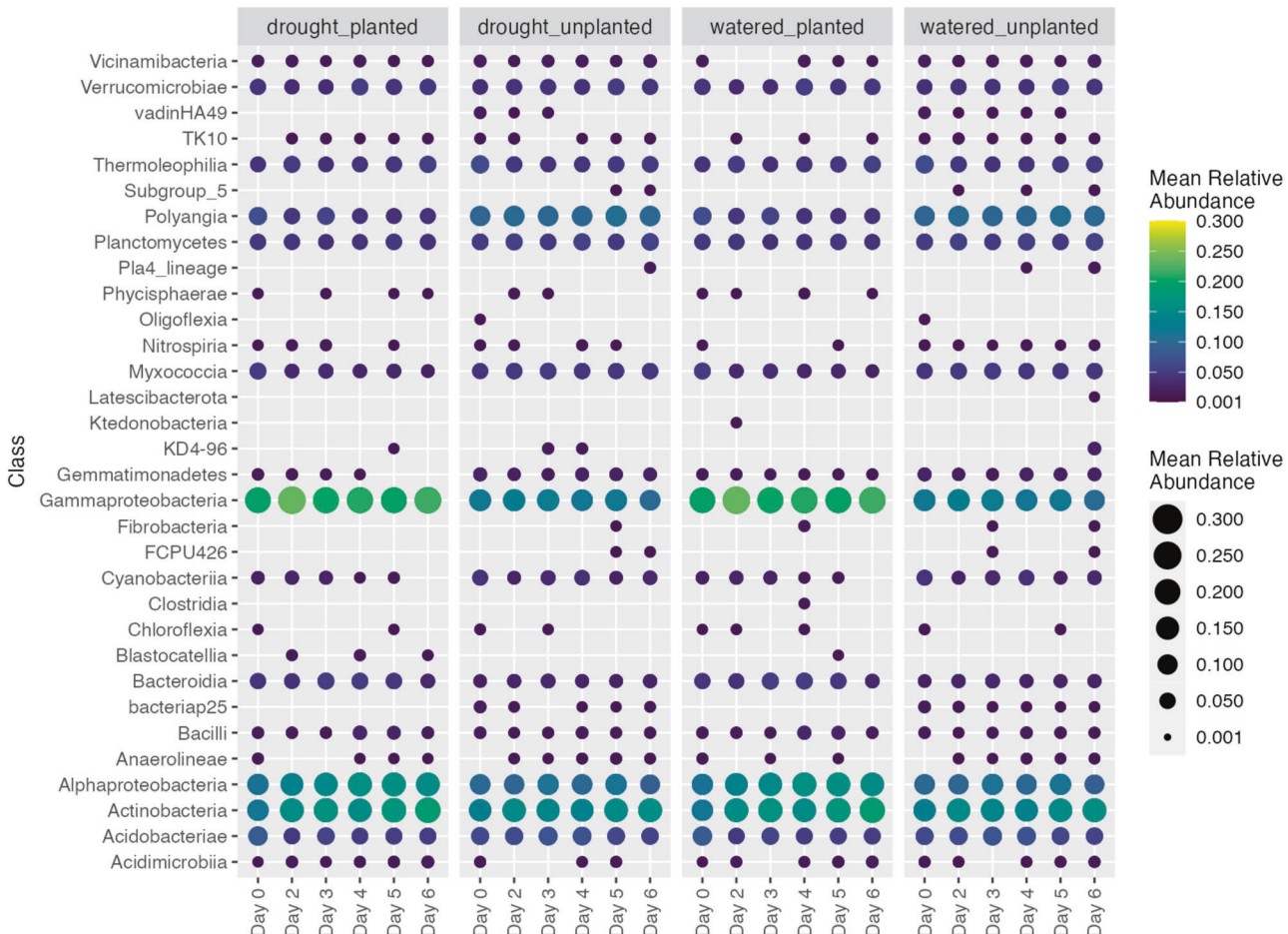

**Fig. 6 | Temporal changes in relative abundances of active taxa that were the most abundant bacterial classes in bean shown across drought/watered and planted/unplanted conditions.** Mean relative abundance of five replicates is plotted per treatment-time condition (sample size (*n*)=105). Source data are provided as a Source Data file.

reduction during drought was greater in bean than switchgrass, which was expected because switchgrass is more tolerant to drought than bean. Related, the switchgrass shoot biomass was less sensitive to drought compared to that of the bean plants.

We deliberately used rhizosphere soils from fields with a recent legacy of growing each crop and thus had an imprint of their typically-associated rhizosphere microbiota. We predicted that in doing so, the microbiome assembly during short-term drought would be biologically relevant to the field and crop. While using different starting soil for each crop does not allow for direct comparison of their compositions, we were primarily interested in the *dynamics* of beta diversity, which we analyzed and compared relative to each soil's Day 0 (pre-drought) time point. As a secondary objective, we were curious if any lineages may be consistently activated in response to drought across the crops, suggesting a general drought response regardless of the plant species. However, the major differences in the active community structures between bean and switchgrass soils were maintained in drought and watered conditions and over time, with no indication of convergence across the two communities given the drought and thus no detection of taxa that may be classified as plant-associated drought generalists. We detected no specific taxa that were generally selected in drought across the different crops, neither at the OTU or family taxonomic level. While other studies have suggested that there could be similarities in the drought response of the microbiome across phylogenetically distant and related host species[47], our study suggests that developing universal, "drought-supportive" targets for microbiome manipulation could instead require solutions tailored to

different plant families. Despite differences in rhizobiome compositions and different host sensitivities to drought, we observed that both bean and switchgrass rhizobiomes changed comparably given water reduction and over time with increasing drought severity. The major difference between the crops was that the presence of a plant was more important for the bean rhizobiome response to drought. At the same time, the plant presence was largely inconsequential for the switchgrass rhizobiome. Generally, switchgrass rhizobiome differences between the planted and unplanted treatments were weak relative to the bean.

For the bean rhizobiomes, several OTUs within classes Actinobacteria and Alphaproteobacteria were enriched in the planted samples compared to unplanted samples during drought. Actinobacteria have generally been seen to be enriched during drought in plant roots[48,49] and are well-known for their versatile capabilities ranging from stress tolerance to bioremediation. We also found that Alphaproteobacteria and Actinobacteria increased in relative abundance over drought severity for bean-planted treatments during drought.

Physiological and morphological differences between bean and switchgrass could explain the differences observed in their assembly of the rhizobiome during drought stress. Switchgrass is known to be generally drought-tolerant[37]. Its extensive and deep root architecture could indicate better resource and water allocation that benefits switchgrass during drought, making it less reliant on beneficial support from the plant's microbiome. On the other hand, bean plants could benefit from directed manipulation of the microbiome during short-term drought, given its wide range of drought susceptibility

across genotypes[39]. A more expansive analysis of the microbiomes of perennial versus annual life histories and among plants with different physiologies could provide insights into expectations of different microbiome drought responses. It could be that the rhizosphere microbiomes of perennial plants are generally more resistant to stress. Notably, in our study, the metabolome results show that the switchgrass plants were responsive to the drought and had clear shifts in some expected features of their metabolite profile as compared to the watered and unplanted conditions. Thus, the metabolite and microbiome changes were not necessarily coupled at the temporal scale of the experiment (days), nor were they collectively indicative of the plant-mediated shift in metabolism that subsequently drove a shift in its rhizosphere microbiome. The observation that the switchgrass microbiome was relatively stable during drought could be due in part to specialized metabolites that switchgrass produces to combat stress.

Saponins are a group of specialized plant metabolites that can exhibit biological activities such as antibacterial, antifungal, and cytotoxic properties[9,50]. Saponins are secreted from plant roots into the rhizosphere and can alter soil bacterial communities[51]. Saponins also have a role in detoxifying free radicals[52] thus helping to combat the oxidative stress imposed by drought. In this study, saponins were increased in concentration in the planted switchgrass rhizospheres under drought. In other studies, saponin concentrations also increased in leaves of different plant species under water-deficit conditions[53,54]. Actinobacteria are known to degrade saponins in the rhizosphere. They are also known as assimilators of saponins, which can explain the co-occurrence of saponins with Actinobacterial OTUs in our study[51]. In addition, the increased accumulations of diterpenoids were also observed in the planted rhizosphere soils stressed by drought. This is consistent with a recent metabolomics study that showed that diterpenoids are also enriched in switchgrass roots after a longer period of drought stress (five weeks)[36].

We identified several bean OTUs that selectively increased with increasing drought severity. This included OTUs belonging to classes Alphaproteobacteria, Blastocatellia, Actinobacteria, Thermoleophilia, Gammaproteobacteria, Verrucomicrobiae, Acidobacteriae, and Acidimicrobiia. These short-term stress responders are critical for further exploration as they may be potential bioinoculant targets.

Our study suggests that considering the active members of the rhizobiome provides complementary insights into the ecology of drought and can suggest different responses of taxa. This study is different in its focus on detecting active taxa by rRNA:rRNA gene ratios. Other studies typically consider in aggregate the total rhizobiome community using DNA-based sequencing, including active, inactive, and deceased members. From these DNA-based studies, the literature reports that soils with limited moisture and under drought conditions have reduced richness of Proteobacteria and Verrucomicrobia[55]. However, in our study, members of these groups were active and abundant in the rhizobiome of drought, unplanted bean soil. Out of the 46 bean taxa uniquely detected in the unplanted drought, several OTUs belonging to these phyla were detected as active, persistent, and in some cases, in relatively high abundance. This included, for example, two Verrucomicrobiae OTUs (OTU#543e7a460e393e133062fbfae2f5675e and OTU#37184df094 89f56490af1670195a8be5) as well as a Gammaproteobacteria OTU (OTU#da195c41827a582bf1b4bd7cb9007452) and Alphaproteobacteria OTU (OTU#fa5570e95914a27603f466f8d4789a26).

Furthermore, though Actinobacteria are widely detected in drought soil[56–58], out of the 46 unique taxa for the unplanted drought samples, we only found one OTU belonging to Actinobacteria (OTU# 78b14ac22ebd6a508a7f122a42beb87a). Interestingly, we did not find this Actinobacteria OTU in the planted drought samples in bean rhizobiomes, though several other OTUs were detected from Actinobacteria. It could be that Actinobacteria are persistent but perhaps dormant or not highly active under some drought conditions and that

specific OTUs have distinct responses during drought treatments that could change under the influence of a plant. Thus, broad generalizations at class or family levels remain challenging and specific cases must be evaluated at deeper taxonomic resolution.

The RNA: DNA ratio method also has limitations that must be carefully considered in interpreting these results. For example, we set a ratio threshold of 1 for the detection of active taxa, which removed several taxa from consideration and, as such, retained only those that met our criteria. Thus, we could be missing the detection of additional active OTUs, especially among rare taxa. Other studies have applied RNA: DNA gene ratios ranging from 0.5-2[28], with five considered overly conservative[21]. However, after the active taxa were designated, it was important to use the DNA sequence counts for taxon relative abundances (and not the RNA counts) due to the expected variation in transcription from taxon to taxon[59].

It was both a feature and a limitation of the study to compare two very different species of plants that have notable differences in many of their traits that could be important for rhizobiome assembly during drought. For example, root biomass, along with differences in root architecture, exudation, total root surface area and other physiological properties of the two crops could help to explain the differences observed in the rhizobiome response to drought. An experimental limitation is that we were not able to assess root biomass due to the destructive sampling necessary for metabolomics and microbiome assessment. Given this, we elected to grow each crop to a stage that maximized the root biomass prior to treatment to ensure maximum comparability in root volume between the plant species. Also, bean and switchgrass likely have differences in their nutrient transportation and exudation profiles. This, along with other plant traits such as life history strategies, likely contributed collectively to the crop-specific microbiome responses to short-term drought. Metabolomics data from switchgrass revealed differences in metabolite profiles between planted and unplanted soils, further suggesting the importance of exudation for rhizobiome recruitment. Given this study's results that suggest there is not a general plant-associated drought microbiome, further work can be done within and across related plant species to understand the plant-driven mechanisms of microbiome activation and deactivation during drought.

Another consideration of this study is the intense and short duration of the drought exposure, accumulating over six days. Our goal was to understand immediate drought response and selection of members in the rhizosphere, so we deliberately chose a shorter timeframe and pulse event to avoid the inclusion of longer-term assembly dynamics. As the rhizobiome assembly continues with drought and re-wetting, its activity dynamics are expected to change. Thus, this work presents the immediate consequences of short-term rhizobiome drought under greenhouse-controlled conditions, and additional work is needed to understand longer-term consequences and how these responses may change given the more complex context in agricultural fields with unpotted plants. Nevertheless, the results from controlled experiments like these, which can specifically manipulate water availability per plant in isolation, are useful to inform the next research steps that could be executed in field studies.

Although this study focused on bacterial community dynamics in the rhizosphere soil during a short-term drought, it is known that the fungal community, particularly the arbuscular mycorrhizal fungi (AMF), can play an important role in drought tolerance of crops[60]. It was beyond the scope of the present study to include both bacterial and AMF characterization, but perennial plants can be more dependent on AMF than annual during drought[61]. Thus, understanding the interplay of AM fungi and the bacterial community for different plant species during drought is a remaining knowledge gap to be filled.

This study assessed the active rhizobiome responses to short-term drought for two different crops: switchgrass and bean. Both plants' active rhizobiomes changed given the drought and over time

with drought intensity, but the response was more muted for switchgrass than for bean. Using unplanted controls, we determined that there were plant-mediated effects of drought on the active rhizobiome only for bean plants and not for switchgrass, suggesting that different plants have different reliance on or engagements with their rhizobiomes during drought. There was unique recruitment of taxa to droughted bean plants, and we could distinguish among the plant- and environment-responsive taxa. This work identifies rhizobiome taxa that may be recruited by or lost by bean plants during drought, which can be targeted to improve understanding and tested for plant benefits.

## Methods

### Experimental design

We performed a greenhouse experiment with common bean (*Phaseolus vulgaris var*. Red Hawk) and switchgrass (*Panicum virgatum var*. Cave-in-Rock). The experimental design consisted of two crop levels (bean, switchgrass), two planting levels (planted, unplanted), two moisture levels (drought, watered), and five destructive plant and rhizosphere sampling points correlating with increased drought severity. The drought gradient included a pre-drought baseline that was watered, and then destructive sampling over the next six days, with the day 2 samples experiencing the lowest drought severity and day 6 samples experiencing the highest. Five replicates were collected at each time point for each combination of the factors tested (Fig. 1).

### Soil sample collection

Bean soil was collected from Montcalm Research Center on September 6, 2018, at Stanton, MI (43.350885, −85.177044), and was most recently planted with common bean (*Phaseolus vulgaris* var. Red Hawk) that year. Switchgrass samples were collected from the Great Lakes Bioenergy Research Center switchgrass plots at Lux Arbor Reserve (42.475224, −85.444979) in Denton, MI, on August 27, 2018. This site has been under continuous switchgrass (*Panicum virgatum* var. Cave-in-rock) cultivation since 2011. Switchgrass soil was collected and homogenized from six sites randomly situated throughout the plots. All sampling materials were sterilized with ethanol before use and between samples. A shovel was used to collect bean and switchgrass soils to a depth of 10 cm. Large rocks and root fragments were removed from the soil. Within three hours (for bean soils) and five hours (for switchgrass soils), soils were transported to the lab and stored at 4 °C until sieving. Soil nutrient analysis was completed for the field soils at the Michigan State University Soil and Plant Nutrient Laboratory (Supplementary Table 4).

Soil samples were sieved (4 mm mesh) over three to five days immediately after the sample collection day and stored at 4 °C until needed for the experiment. The drought experiment commenced after eight months and six months of soil storage at 4 °C, for bean and switchgrass, respectively.

### Seed germination

Approximately 250 switchgrass seeds (Sharp Bros. Seed of Mo., Inc) were placed in 50 ml conical tubes. 10 ml of 5% bleach was added to the tubes. The tubes were placed on a shaker at 200 rpm for 20 min. The bleach/water mix was decanted, and 10 ml of deionized water was added. The tubes were agitated again at 200 rpm for 5 min, then decanted. The deionized water rinse was repeated two more times. Approximately 60 switchgrass seeds were placed onto a wet filter paper (90 mm diameter) per sterile Petri plate. Seeds were moistened with deionized water, plates were wrapped in parafilm, covered with foil to block light, and incubated at 32 °C incubator for three days. After three days, 1 ml of deionized water was supplemented to each plate, which was re-wrapped and returned to 32 °C for an additional day.

Bean seeds were obtained from the Dry Bean Breeding and Genetics program at Michigan State University. Approximately 590 seeds were placed in a 1 L Erlenmeyer flask, and 1 L of sterilization solution was added (0.1% Tween-20, 10% bleach, and 90% DI water) to the seeds. The seeds were soaked for 15 min at room temperature and inverted every few minutes. The seeds were washed five times with deionized water, and 12 seeds were placed onto Petri dishes lined with moistened filter paper. Plates were wrapped in parafilm and incubated at room temperature in the dark. Over the next three days, 2 ml of Milli-q water was added to each plate. After four days, approximately 60% of the switchgrass and bean seeds germinated and had an emerged radicle, and these were advanced for use in the experiment.

### Growth tube preparation and planting

Growth tubes were made by modifying 50 ml conical tubes to better control soil water content during the experiment. An ethanol-cleaned drill bit was used to drill a hole in the bottom of 50-ml conical tubes. A hole was also drilled in every cell of the tube racks used to hold the 50 ml tubes for switchgrass and bean plants. Small squares of Kimwipe® tissue papers (5 cm * 5 cm) were cut, each rolled, and one rolled sheet was used to plug the drain hole in each 50 ml conical tube (Supplementary Fig. 1). Autoclaved perlite was combined with sieved field soil (50:50 v/v) to fill the prepared tubes to the top. Soils were watered with 10 ml of deionized water, which reduced the total soil volume to approximately 35 ml. Tubes for the planted treatment received one germinated seed. Unplanted tubes were prepared the same but without a seed. Seeds were planted so that ~1 cm of the seedling was exposed above the soil and at a depth to ensure that the emerging root would not push the seedling out of the soil. After planting, additional soil: perlite mix was added to the tube to replace the volume lost after settling and supplemented with 3 ml of deionized water. Each tube was wrapped in foil to exclude light in the root zone. The greenhouse's daily temperature range was 21-32 °C, with 14 h of supplemental daytime lighting (400-watt high-pressure sodium lamp).

Seedlings were provided sufficient water to achieve healthy growth before initiating the drought. Switchgrass tubes were watered with 10 ml of deionized water every other day or every two days for two months until the plants were large enough for the experiment, which was designated as when the roots had filled the tube. Though this was close to the flowering stage, no flowers were observed. Bean tubes were watered daily with 10 ml of deionized water for the first five days and then twice daily with 10 ml for four additional days. After ~9 days of watered conditions, approximately 55 bean plants were sorted by height as a proxy for total biomass, and similarly sized plants were selected to include in the experiment. At this vegetative stage, the first trifoliate leaf emerged and started to unfold in all bean plants, and roots were completely bound within the tube. Before the drought, the planted tubes of both switchgrass and bean had tube-bound root systems such that all their soil could be directly influenced by the plant and classified as the rhizosphere. Bean roots were tube-bound within a week, but switchgrass took about two months. The timing for the start of drought treatment was chosen based on the time needed for the roots to fill the tube so that the entire soil was plant-influenced rhizosphere. Thus, the growth stage at the time of drought commencement was different between the two crops.

### Drought treatment and destructive sampling

The same experimental design and sampling strategy were applied to bean and switchgrass (Fig. 1). Twenty-five seedlings and 25 unplanted tubes were assigned to either the sufficiently watered ("watered") or to the reduced water treatment ("drought"), totaling 100 samples. In addition, five planted and five unplanted tubes were collected before the drought ("pre-drought," day 0). These samples were used as a baseline control to compare to the subsequent drought samples. The drought was initiated on the same day the pre-drought samples were

collected (see Supplementary Tables 5 and 6 for the watering regimes for switchgrass and bean, respectively). The drought severity was increased over six days by adding progressively less water. Five replicate soil samples (individual tubes used for destructive sampling) were collected for each experimental condition from day 2 to day 6 for five post-drought sampling points.

To ensure that the drought bean plants remained viable (and that our assessment was of the rhizobiome response to live plants rather than to dead), an additional 36 plants were in parallel subjected to drought to determine their viability on the final days of the experiment. On day 5, when bean plants were browning and lacked turgor (day 5), half (18) of the extra plants were provided 15 ml of deionized water in the morning and the afternoon. All 18 plants were resuscitated, and we observed their leaves returned to full turgor by the next morning. Thus, the drought plants from day 5 were likely alive when sampled despite appearing well-desiccated. The same was repeated on day 6 with the remaining drought plants. Thirteen of the 18 plants resuscitated (leaves back to full turgor) within two days, affirming that most drought plants on day 6 were viable. As a relatively more drought-tolerant plant, switchgrass did not appear severely desiccated on days 5 and 6, and the resuscitation check was unnecessary to confirm its viability.

### Soil, root, and shoot sample collection and gravimetric soil moisture

Materials and surfaces were sterilized and/or ethanol-cleaned before use and between samples. All tubes were destructively sampled. During each sampling, each tube's contents were dumped onto ethanol-cleaned aluminum foil. The soil was brushed from the root system using ethanol-cleaned gloves and homogenized with an ethanol-cleaned metal spatula. Gloves were ethanol-cleaned again when handling new samples and new ethanol-cleaned gloves were used as needed. Roots were manually removed from the soil, and 0.5 g of soil was collected into a 1.5 ml microcentrifuge tube for DNA/RNA co-extractions. An additional ~6 g soil was retained as a backup. Perlite (>2 mm) was avoided. Soils were flash-frozen in liquid nitrogen and stored at −80 °C until nucleic acid extractions. An additional 4 g of soil was collected to determine gravimetric soil moisture content. After recording the mass of fresh soil collected, soils were dried in an oven at 50 °C for at least three days and then re-weighed. Percent gravimetric soil moisture was calculated as the percent water mass lost during drying using Eq. 1[62].

$$Gravimetric\ moisture\ content(\%) = \frac{(Mass\ of\ soil_{wet} - Mass\ of\ soil_{dry})}{Mass\ of\ soil_{dry}} *100$$

(1)

After soil collection, the shoot system was cut at the base of the stem and put in pre-labeled envelopes. The shoots were dried at 50 °C for at least five days. Masses taken after drying were used to determine the biomass of the shoot. Treatment differences in shoot biomass and gravimetric soil moisture were analyzed with a three-way ANOVA type III test upon satisfaction of normality.

An additional 30 switchgrass plants (15 watered and 15 drought) were grown for metabolite analysis on day 6. Soils were homogenized, shoots were sampled as previously described, and root systems were cleaned in deionized water. Then, samples were flash-frozen in liquid nitrogen in 15 ml tubes. Soils of the unplanted treatments (watered and drought) from day 6, stored at −80 °C, were used to compare with the planted treatments. We also analyzed metabolites from pre-drought soils (four replicates of planted and five unplanted treatments) stored at −80 °C. In addition, ten unplanted soils from day 6 (five drought and five watered) and nine pre-drought samples (four planted, five unplanted) were analyzed for metabolites. In summary, 15 switchgrass

plants from the day 6 watered and drought treatments each yielded 30 samples for soil, shoot, and root metabolites.

### Soil DNA/RNA co-extractions

DNA/RNA co-extractions were performed using the protocol specified in ref. 63 with minor modifications. Briefly, 0.5 g of flash-frozen soil was added to 0.7 mm PowerBead® garnet bead tubes (Qiagen, Germantown, MD, purchased early 2021 before being discontinued). 0.5 ml of CTAB-phosphate buffer (120 mM, pH 8) and 0.5 ml of phenol:chloroform: isoamyl alcohol solution (25:24:1, nuclease-free, Invitrogen®) were added to the tube and placed on a bead beater to beat for 30 s. Tubes were then centrifuged at 12000 g for 10 min at 4 °C. The top aqueous layer was extracted and placed into a new 1.5 ml microcentrifuge tube. Next, 0.5 ml of chloroform: isoamyl alcohol (24:1, nuclease-free, Sigma®) was added to the tube and inverted several times to form an emulsion to remove residual phenol. Tubes were centrifuged at 16000 g for 5 min at 4 °C, the top aqueous layer extracted and placed into a new 1.5 ml microcentrifuge tube. Nucleic acids were precipitated by adding two volumes of 30% polyethylene glycol solution (PEG6000, 1.6 M NaCl) and mixing a few times. Tubes were incubated on ice for two hours. Tubes were then centrifuged at 16000 g for 20 min at 4 °C. The supernatant was removed, and 1 ml of 70% ice-cold ethanol was added. Tubes were centrifuged at 16000 g for 15 min at 4 °C. Ethanol wash was pipetted out, being careful not to remove the pellet, and placed back in the centrifuge for a final spin for 10 s to collect residual ethanol. The remaining ethanol was pipetted out of tubes, and the pellet was allowed to air dry. The nucleic acid pellet was resuspended in 30 μl of nuclease-free water. Negative controls for extractions included tubes to which no soil sample was added, and only included reagents and the garnet beads to check contamination in the extraction reagents. Two negative controls were processed alongside the experimental samples for each extraction day. All DNA and RNA samples were quantified using Qubit® dsDNA BR assay kit and RNA HS assay kit on a qubit 2.0 fluorometer (Invitrogen, Carlsbad, CA, USA). All nucleic acid raw coextracts (containing DNA and RNA) were visualized using agarose gel electrophoresis and validated with a band for DNA and RNA[63].

### DNase treatment and cDNA synthesis

The DNA/RNA coextract was used to prepare purified RNA using the Invitrogen TURBO® DNA-free Kit with minor modifications. 1 μl of 10X TURBO® DNase Buffer and 3 μl of TURBO® DNase enzyme were added to a 6 μl aliquot of the DNA/RNA coextract. The mixture was then incubated for 30 mins at 37 °C, after which 2 μl of a DNAse inactivation reagent was added to each tube. The resulting solution was mixed by flicking the tubes by hand and then incubated at room temperature for 5 mins. The samples were centrifuged at 2000g for 5 min at room temperature. The purified, DNAse-treated RNA samples were transferred to a clean, sterile tube and immediately processed to make complementary DNA (cDNA). cDNA was prepared using the Invitrogen® SuperScript III First Strand Synthesis System using random hexamers per the manufacturer's instructions. cDNA samples were stored at −20 °C until further processing. Negative controls were prepared for the cDNA synthesis step to check for reagent contamination.

### PCR and RT-PCR

DNA (PCR) and cDNA samples (RT-PCR) were amplified using a standard protocol specified by the Research Technology Support Facility at Michigan State University. A 15 μl reaction volume was prepared for each sample to amplify the V4 hypervariable region of the 16S rRNA gene with 2X GoTaq Green Mastermix, primers 515 F (5′- GTGC CAGCMGCCGCGGTAA- 3′) and 806 R (5′-GGACTACHVGGGTWTCT AAT-3′)[64] and 1 μl template. The final concentration of the GoTaq Green Mastermix was 1X, and the final concentration of each primer was 0.1 μM. The PCR cycle was run using the following cycling

parameters: initial denaturation at 94 °C for 3 min; 30 cycles of denaturation at 94 °C for 45 s, annealing at 50 °C for 60 s, and extension at 72 °C for 90 s, followed by a final extension at 72 °C for 10 min. Samples were kept at 4 °C and immediately visualized on a 1% agarose gel using 100-bp ladder. All RNA samples purified from DNA/RNA coextracts and used to prepare cDNA were also used to run PCR and check on 1% agarose gel to ensure there were no contaminating DNA bands in RNA samples. All PCR reactions included a no-template negative control and an *E. coli* DNA template as a positive control. All negative controls used for extractions were also included as samples in the PCRs. Results indicated no contamination from extraction or PCR reagents and consistent amplification performance with the cycling parameters.

## Illumina sequencing of the 16S rRNA gene and 16S rRNA

Amplicon sequencing was performed at the Genomics Research Technology Support Facility at Michigan State University. The V4 hypervariable region of the 16S rRNA gene was amplified using dual-indexed Illumina compatible primers 515F and 806R as described in ref. [65]. PCR products were batch normalized using an Invitrogen SequalPrep DNA Normalization Plate, and the normalized products recovered from each of the six plates submitted were pooled. The pools were cleaned up and concentrated using a QIAquick PCR Purification column followed by AMPureXP magnetic beads; it was quality controlled and quantified using a combination of Qubit dsDNA HS, Agilent 4200 TapeStation HS DNA1000, and Invitrogen Collibri Library Quantification qPCR assays.

Each plate submitted for sequencing had approximately 85 samples (6 plates); each plate was used for one MiSeq run. The pools from each plate were each loaded onto an Illumina MiSeq v2 standard flow cell, and sequencing was performed in a 2 x 250bp paired-end format using a MiSeq v2 500 cycle reagent cartridge. Custom Sequencing and index primers were added to appropriate wells of the reagent cartridge. Base calling was done by Illumina Real Time Analysis (RTA) v1.18.54, and the output of RTA was demultiplexed and converted to FastQ format with Illumina Bcl2fastq v2.20.0. In addition to experimental samples, all negative controls from each extraction day were sequenced, and these samples were evenly distributed throughout all the MiSeq runs. Two positive controls (mock communities) were sequenced with each MiSeq run. One positive control was an in-house Mock community prepared in the Shade Lab[66], and the other positive control was provided by the RTSF at MSU for library preparation.

## Metabolite extraction from plant tissue and soil

The switchgrass shoot, root, and rhizosphere soil samples collected from 15 individual plants in the drought experiment were pooled into seven replicates (Supplementary Fig. 7A, B). We pooled plant tissue samples to achieve the 0.1-0. 5 g of dry plant tissue required to extract the total metabolites for metabolomics analysis. To ensure enough mass for extraction and appropriate replication for statistical analysis, we divided the 15 individual plants into 7 groups: 6 groups included 2 plants and the last group included 3 plants. After grouping, the tissues were ground into powders and metabolites were then extracted from an equivalent mass per pool. Though the soil samples did not require pooling to achieve sufficient mass, we performed the same grouping and extraction protocols to the soils to maintain consistency.

The plant tissues were lyophilized and ground into powders using an automated tissue homogenizer (SPEX SamplePrep, Metuchen, NJ). The plant metabolites were extracted from the powders using 80% methanol containing 1 μM telmisartan internal standard and normalized by tissue weight to achieve equal concentration as described in ref. 8. The soil metabolites were extracted following the same protocol with differences in the soil-to-solvent ratio (1:2, v/v) and incubation method (mixing on a laboratory tube rocker). Extracts were centrifuged at 4000 g for 20 min at room temperature to remove solids. The supernatant was completely dried using a SpeedVac vacuum

concentrator (ThermoFisher, Waltham, MA) and reconstituted using 1/10 the original solvent volume.

## Liquid chromatography–mass spectrometry (LC-MS) based untargeted metabolomics

The chromatographic separation and MS analysis for the switchgrass metabolites were performed using a reversed-phase, UPLC BEH C18, 2.1 mm × 150 mm, 1.7 μm column (Waters, Milford, MA) and an Electrospray Ionization – Quadrupole Time-of-Flight MS (ESI-QToF-MS, Waters). Mass spectra, under positive ionization, were acquired by data-independent acquisition (DIA)as described in Li, et al.[8]. The untargeted metabolomics data processing, including retention time (RT) alignment, lock mass correction, peak detection, adduct grouping and deconvolution, and metabolite annotation, were done using the Progenesis QI software package (v3.0, Waters) following the protocol in Li, et al.[8]. The identified analytical signals were defined by the RT and mass-to-charge ratio ($m/z$) information and referred to as the *features*. Measurement of each feature across the sample panel was filtered by interquartile range, log-transformed, and scaled for multivariate analyses using R Studio v3.1.1 and R package MetabolAnalyze (scaling function set to type "pareto"). The log-transformed and pareto scaled (normalized) abundance of features for soil were used for principal component analysis (PCA). A Partial Least-Squares Discriminant Analysis (PLS-DA) model was generated using the MetaboAnalyst 4.0 online tool platform[67] to assess each metabolite feature's variable of importance (VIP) coefficient. The top 50 features with the highest VIP coefficient were used to visualize in a heatmap. For this, the features were log-transformed and scaled from 0-1 (by maximum) across all the samples. This was completed using R software and then visualized using the heatmap function in the R package ComplexHeatmap. Distance and clustering methods were set to Euclidean and Ward.D to generate hierarchical clustering for the heatmap. The data dependent acquisition (DDA) was done only for one pooled sample in order to obtain MS/MS spectra for annotating the top 50 PLS-DA features using CANOPUS (class assignment and ontology prediction using mass spectrometry)[68] machine learning function built in the SIRIUS 4 (https://bio.informatik.uni-jena.de/sirius/), a computational tool for systematic compound class annotation. The identification level was denoted for each annotated metabolite based on the criteria for metabolite identification as per Sumner, et al.[69].

## 16S rRNA and 16S rRNA gene sequence analysis

Sequence data were analyzed using QIIME2[70]. All paired-end sequences with quality scores were compressed and denoised using the DADA2 plugin[71]. The denoising step dereplicated sequences, filtered chimeras, and merged paired-end reads. The truncation parameters to use with the DADA2 plugin were determined using FIGARO[72]. FIGARO analyzes error rates in a directory of FASTQ files to determine the optimal trimming parameters for sequencing pipelines that utilize DADA2. The truncation length was set to 123F and 162R for all the data, with minimum overlap set to 30 base pairs, which resulted in 93% merging success. All truncation was performed from the 3' end for consistent final read lengths. The DNA and cDNA datasets were separately denoised. The resulting DNA and cDNA count tables were merged into a single QIIME2 artifact using the feature-table merge command.

Similarly, the DNA and cDNA representative sequences were merged into a single QIIME2 artifact using the feature-table merge-seqs command. The representative sequences from the combined count tables were clustered at 99% identity de-novo, and the clustered representative sequences were classified using SILVA v138[73] to generate the taxonomy file. Ninety-nine percent sequence identity was used to define OTUs to conservatively account for any potential amplification errors that may have occurred during the cDNA synthesis from the RNA. The resulting OTU (operational taxonomic unit) table and taxonomy files were exported to R for ecological analysis.

## Designating the active community members

All downstream analyses were performed in R version 4.0. The R package decontam[74] was used to determine the number and identity of contaminants in the dataset (Supplementary Fig. 8A and 8B) and remove them using the prevalence method. Contaminating taxa, mitochondria, and chloroplast sequences were filtered from the datasets. Based on rarefaction curves, a subsampling depth of 15,000 reads per sample was selected (Supplementary Fig. 8C). After subsampling, 16S rRNA to rRNA gene ratios (hereafter, 16S rRNA:rRNA gene) were computed from the DNA and cDNA datasets as described in ref. 21. While we compared a few methods therein for this dataset (please see Supplemental Information for details), we ultimately chose the method that applied a 16S rRNA:rRNA gene ratio threshold > =1. The chosen method was statistically robust in overarching patterns of beta-diversity given the exclusion or inclusion of phantom taxa (taxa with detected cDNA but not DNA counts). For phantom taxa that were detected in greater than 5% of samples, the DNA counts = 0 were changed to DNA = =1 (as in "method 2" in ref. 21) (see Supplemental Materials). All other phantom taxa were excluded. The DNA OTU table was filtered to include only sequence counts of active taxa in the samples determined to meet our ratio threshold. Consequently, while every DNA and cDNA sample of sequence counts was initially rarefied to 15,000 reads, each sample's active community varied in their total reads (2000–6000). Relativized abundances were used for ecological statistics.

## Microbiome data analysis for active community

The OTU table of the DNA counts of active taxa, taxonomy table, and metadata files was merged using the phyloseq package[75] in R. We used Bray-Curtis dissimilarity to determine beta diversity but also tested weighted UniFrac distance, and results were comparable. Permutational analysis of variance (PERMANOVA) was conducted using the adonis function in vegan package[76] to assess differences in community structure by treatment and interactions: drought treatment (watered or drought), drought severity/sampling (days 2-6), and plant treatment (planted or unplanted). For post hoc tests, pairwise comparisons between drought levels were computed using the pairwise.adonis2 function in vegan package. PERMANOVA tests using the adonis function in R vegan package were also done on metabolite feature abundances used for multivariate analysis to understand the effects of planting, drought, and sampling day factors. To understand drought dynamics, we analyzed the Bray Curtis similarity of the microbiome to the pre-drought samples over the covariate of time for planted and unplanted treatments in bean and switchgrass. This was visualized using a smoothed conditional means (geom_smooth function) with a linear model. This same approach was also used to assess general, relative fluctuations in rhizobiome size across samples, as proxied by DNA concentration.

Richness was the observed number of OTUs, using the estimate_richness function from 100 re-samples of the community. The normality of alpha diversity metrics was assessed using a Shapiro-Wilk test with a cut-off of W > 0.9 for normality assumptions. Since data were normally distributed, a parametric three-way ANOVA test was used to assess the main effects and interaction effects between factors on richness. Contrasts were set to "sum" before running the ANOVA model to ensure that type III ANOVA tests were valid. We used non-parametric Kruskal-Wallis tests (Kruskal.test in R) to calculate differences in class-level relative abundances between treatments.

We used the indicspecies package in R to determine indicator species associated with experimental conditions. We used the abundance-based counterpart of Pearson's phi coefficient of association within the multipatt function[77]. To correct the phi coefficient for unequal group sizes the "func" parameter within multipatt was set to "r.g." P values were adjusted for false discovery rates. Indicator species

were calculated at the OTU level for each treatment combination and the OTU and family level for the crop-specific indicators.

We created a heatmap visualization of the 50 most abundant and active OTUs in the bean dataset for indicator species. We used a maximum standardization approach using the decostand function in the vegan package in R. We distinguished taxa that changed relative abundance over the drought gradient from taxa that changed in their detection. For samples without detection of an OTU's activity or DNA (e.g., DNA = 0, cDNA = 0), that OTU's abundance was coded as NA. For samples that had no detection of an OTU's activity but had detection of its DNA (e.g., DNA > 0, cDNA = 0), that OTU's abundance was coded as 0. For samples that detected an OTU's activity and DNA (e.g., DNA > 0, cDNA >0), that OTU's abundance was the value of its DNA sequence count. We also included a special consideration of "phantom" OTUs (e.g., DNA = 0, cDNA>0), which were assigned a sequence count of 1.

## Statistics and Reproducibility

In summary, we conducted a greenhouse experiment to understand active bacterial community dynamics during short-term drought. We used field soil with a legacy of growing two crops, switchgrass and common bean and conducted a drought treatment exposure over 6 days with incremental drought severity over time. We used watered controls and a pre-drought baseline for all comparisons. Appropriate multivariate statistical tests were conducted on the rhizobiome and metabolomics data, along with statistical tests for univariate data such as gravimetric moisture, shoot biomass and DNA biomass yield. Details are provided in the Methods sections above. All greenhouse experiments were conducted at least twice, which included a pilot study, which dictated the water addition levels appropriate to inflict drought in the respective crops without affecting the survival of those crops. Treatments were assigned randomly to crops during the greenhouse experiment. Even though no statistical method was used to predetermine sample size, we opted for 5 replicates as they accurately captured any differences observed in plant physiology during the experiment. A few data points were excluded from the rhizobiome analyses if they did not pass the rarefaction threshold.

## Reporting summary

Further information on research design is available in the Nature Portfolio Reporting Summary linked to this article.

## Data availability

The sequence data generated in this study has been deposited to NCBI Sequence Read Archive under BioProject number PRJNA862978. The LC-MS metabolomics datasets generated in this study are available on Dryad - 'Metabolomics data for Disentangling plant- and environment-mediated drivers of active rhizosphere bacterial community dynamics during short-term drought'. It has been assigned a digital object identifier (DOI): https://doi.org/10.5061/dryad.6t1g1jx5z. Access to the data files can be obtained using this URL: https://datadryad.org/stash/share/vkqIFDwKkWd0_U36mKWee1z1-pb3Xha3ajrZ7x_HO2c. All raw data files used to generate figures throughout the manuscript have been submitted to Figshare and can be accessed at this https://doi.org/10.6084/m9.figshare.25214351.v1. Source data are provided with this paper.

## Code availability

Scripts for analyzing the microbial community for this study can be found on GitHub at https://github.com/ShadeLab/PAPER_DroughtRhizobiome_Bandopadhyay_2023[78].

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

## Acknowledgements

This work was supported by the National Science Foundation Award Number MCB 1817377 to AS and by the Great Lakes Bioenergy Research Center, U.S. Department of Energy, Office of Science, Office of Biological and Environmental Research under Award Number DE-SC0018409 to Timothy Donohue, and by the National Science Foundation Long-Term Ecological Research Program (DEB 1832042 to Nicholas Haddad) at the Kellogg Biological Station. AS acknowledges salary support from the USDA National Institute of Food and Agriculture and Michigan State University AgBioResearch and from the Centre National de la Recherche Scientifique (CNRS), France. The authors acknowledge the Great Lakes

Bioenergy Research Center Communications team for graphic design used in Fig. 1.

## Author contributions
A.S. and A.W.B. designed the experiment. S.B., X.L., A.W.B. conducted experiments. S.B., X.L., and A.S. generated figures. S.B., X.L., R.L.L. and A.S. analyzed the data. S.B. prepared the first draft. All authors read and approved the final manuscript.

## Competing interests
The authors declare no competing interests.
