## [Peer Review file · Nature Communications]

REVIEWER COMMENTS

Reviewer #1 (Remarks to the Author):

Bandopadhyay et al describe a well-designed study investigating the effects of drought on two crops. Given climate change, understanding how to harness microbiomes to mitigate drought stress is an important research area. Overall, the manuscript is well written with clear figures and text. They provide evidence that active microbes change progressively with drought intensity and that the presence of a plant mediates the responses of the active rhizobiome to drought. These are both interesting results. In terms of making comparisons between crops, e.g. "presence of a plant more strongly explained the rhizobiome variation in bean (17%) than in switchgrass (3%)", I do worry that much of what is being observed has to do with the dramatic differences in root biomass vs. drought responses, exudate composition, etc.

Major points

Did I miss the data on the root biomass weights? How much of the differences observed between the two crops is explained by root biomass? Switchgrass has 1/10 the shoot biomass of the beans and thus would be expected to have much less transportation and exudation. Differences in crop biomass is likely also reflected in the soil moisture analyses which show that the soils in the tubes with the beans have much less soil moisture (the 'drought' is more severe).

I find headings like "the most striking difference between the rhizobiomes of the two crops was whether a plant was present" confusing because the rhizobiome is a rhizosphere community. I assume you are comparing the abundance of the microbes in the rhizosphere with their abundance in the soil and suggest that you clarify this because it's not clear after reading the methods. Again the observation that there is a much smaller influence of the plant with SG could very well be due to the much lower root biomass.

Some discussion should be included on the potential bias introduced by using different soils to grow the two crops.

Make sure you have controlled for multiple comparisons. My understanding is that PERMANOVA does not control for false discovery when doing multiple comparisons and that it is recommended to use the Benjamini-Hochberg procedure.

Minor points

Line 48. A bit unclear what you are comparing here...how do you have a rhizobiome without a plant?

Line 136. More important vs. what? Dormant microbes wouldn't be expected to contribute much

Line 155. Is Red Hawk a drought sensitive line?

Suggest adding the ages of the plants to Figure 1. It's not obvious from the methods.

Was the brush used to remove the soil from the roots sterilized between plants or were new brushes used, should be clarified in the methods.

Line 573. Suggest adding a reference supporting using NDA concentration as a proxy for cell size.

The metabolomics analyses generally look good.

The pooling of the metabolomics samples is confusing, why did you pool samples? Why include one replicate with 3 pooled samples? How did you account for this in the statistical tests?

Line 413. A bit confusing, I assume you scaled each metabolite feature by the maximum across all samples?

The metabolomics data must be deposited in a public repository. I suggest using GNPS because it will provide matches against real molecules and would be a nice complement to the CANOPUS and SIRIUS based annotations.

I'm only seeing feature annotations for the 50 features from the VIP analysis. Where are the CANOPUS/SIRIUS IDs for the rest of the features?

Double check that the 50 features from the VIP analysis are indeed unique features. For example, 10.52_678.3852n and 10.76_678.3851n seem like they could be resulting from the feature finding algorithm. Are they baseline resolved? When you sort the 50 by retention time you see that many of them are very close in RT.

The MS/MS for the 10.11_1000.5271n is very messy and does not provide strong evidence, the text should mention that the MS/MS spectra is low intensity but suggests that this may be the same ion detected in the previous study. Note that S14 from that previous study annotates the feature as "1 x

hexose, 2 x deoxyhexose, 1 x pentose” so I think the name mono-glycosylated is a bit confusing (though this is likely used to indicate that it is glycosylated at one position). Suggest ‘glycosylated steroidal saponin’.

Reviewer #2 (Remarks to the Author):

Bandopadhyay et al investigated bacterial community and plant root metabolites in soils planted or unplanted by bean or switchgrass during a course of short-term drought. Effect of plant presence, plant identity, and drought on bacterial community is detected. Effect of drought on metabolite is also detected. Overall, this reviewer thinks the manuscript is not ready for publication in Nat Comm. Most of the analysis are descriptive and superficial, and the rationale for the experiment design is not well-documented. The results of metabolites and bacterial community remain independent from each other.

Fig. 1 The experiment design is not clearly described in the figure legend. Why metabolomics sampled for switchgrass, but viability assays for bean?

Fig. 3 What is VIP. What’s PLS-DA? Figure legend should be self explainable.

Line 48-50. However, the presence of a plant more strongly explained the rhizobiome variation in bean (17%) than in switchgrass (3%), with a small effect of plant mediation during drought only observed for the bean rhizobiome. May the different of bean and switchgrass due to the difference in the moisture between the bean and switchgrass treatments. As the figure 2 showed that the measured soil moisture varied between bean and switchgrass.

Line 45-46. there were differences in the active microbiome structure between drought and watered and between planted and unplanted treatments. This is already known by a lot of previous studies, and this study provided no novelty.

Despite their different community structures, the drought rhizobiome dynamics were similar across the two crops. Not clear about how which results support this conclusion?

Line 54 its risky to conclude ‘decoupling of plant exudation and rhizobiome responses’.

Fig. 4 Fig. 1 is hard to read. This reviewer suggest not to use different size to depict harvest time. Why the samples of beans formed two cluster?

Fig. 5-6 Why bean only? No switchgrass?

Fig. S7, low resolution, hard to read, what’s the different bars at the horizontal axis?

Reviewer #3 (Remarks to the Author):

General Comments:

In this study, the authors conducted greenhouse studies of common bean and switchgrass to evaluate the bacterial rhizobiome responses to short-term drought (6 day) using 16S rRNA and 16S rRNA gene sequencing to determine if 1) active rhizobiomes respond to drought and change progressively with drought severity and/or 2) the host plant mediates the responses of the active rhizobiome to drought via compositional changes in root metabolites. In addition, metabolite data was collected for switchgrass and plant performance for bean (switchgrass did not demonstrate differences for plant performance for the short-term drought plants compared to watered). Rhizobiome community analysis for the 'active' fraction of the community (OTUs for which their 16SrRNA:rRNA gene ≥ 1) showed that there were statistically supported differences in community structure with drought and time/drought severity for both switchgrass and bean, but that statistical difference in temporal dynamics was not observed. Therefore, the authors determined that hypothesis 1 had been supported by their data. With regard to hypothesis 2, the authors found that both plants influence the rhizobiome, but the bean rhizobiome showed greater response to the presence of the plant (17% explanatory value) compared to switchgrass (3% explanatory value). However, they found that only the bean rhizobiome had a significant structure change in response to drought. Thus, the authors indicated that hypothesis 2 was only partially supported by their data. Because the bean rhizosphere did appear to change in response to drought, the most abundant activated taxa that were uniquely associated with the planted drought condition in bean rhizobiomes were also identified (87 taxa total). Overall this is a well-designed and written study that describes how the switchgrass and bean rhizobiomes respond to a short duration drought under greenhouse-grown conditions. However, the manuscript can be improved with attention to the following items described below.

Specific Comments for the Authors' Consideration:

- 1) The authors designed the plant growth set-up such that the plants were root-bound by the start of applying drought conditions so that it is assumed that all the soil in the pots is in close proximity to soil and can be considered rhizosphere soil. While this setup does simplify root sampling, it is known that root architecture and spatial arrangement does impact rhizobione recruitment and maintenance, so the concern then is, how well does the data generated in this greenhouse study

reflect rhizobiome dynamics that would occur in a 6 day-drought scenario in the field for switchgrass and common bean? The authors should address this concern in their discussion.

2) The authors explain that they used a 6-day drought because they were interested in understanding how rhizobiome communities shift under a short duration drought scenario for a relatively drought resilient (switchgrass) and more drought sensitive plant (common bean) to better understand initial and short-term community changes. The authors should include information from other studies (if available) about how switchgrass and common bean have been shown to respond to short versus longer term drought to provide further rationale for why the 6 day drought period was used.

3) The authors do include commentary on why they relied on 16S rRNA and 16S rRNA gene sequencing to examine rhizobiome changes in response to drought and what the limitations of that methodology can be. I think the authors did develop solid, interpretable data using this method; however, had the authors used a genome-resolved metagenomics analysis approach, then it would be likely that rhizobiome metabolisms that play a role in a productive drought response could be identified, and this would have greater impact to the field.

4) It is not clear from the study design and discussion why the examination of the fungal rhizobiome community was not also included in the analysis since it is known that fungi (especially AM fungi) contribute to the drought response in switchgrass and bean. The authors should also address this limitation in their study discussion.

RESPONSE TO REVIEWER COMMENTS

We thank the reviewers and the editor for their interest in our work and their helpful comments and suggestions for improving this manuscript. We have addressed each reviewer comment below with page numbers referencing changes and additions, and **we annotate each numbered reviewer request as a “comment” in the tracked changes word document provided.**

Reviewer #1 (Remarks to the Author):

R1.1. Bandopadhyay et al describe a well-designed study investigating the effects of drought on two crops. Given climate change, understanding how to harness microbiomes to mitigate drought stress is an important research area. Overall, the manuscript is well written with clear figures and text. They provide evidence that active microbes change progressively with drought intensity and that the presence of a plant mediates the responses of the active rhizobiome to drought. These are both interesting results.

>>> Thank you so much for your time reviewing the manuscript. We are glad that you find the manuscript to be well written with clear figures and text. We are also glad that you agree with the evidence we provided on the active microbial community changing progressively with drought intensity and how a plant mediates the response of the active rhizobiome to drought. Thank you very much for the positive feedback.

R1.2 In terms of making comparisons between crops, e.g. "presence of a plant more strongly explained the rhizobiome variation in bean (17%) than in switchgrass (3%)", I do worry that much of what is being observed has to do with the dramatic differences in root biomass vs. drought responses, exudate composition, etc.

>>> Thank you for pointing out the importance of root architecture and root biomass. We agree that the precise reason for the plant-to-plant differences in drought response is not knowable from this experimental design, and they could be due to differences in exudation or architecture or other physiological property of the two species. We agree that we cannot speculate beyond reporting that there were plant-to-plant differences.

Because the metabolomics (and microbiome) protocols are necessarily destructive, we could not assess both root biomass and metabolomics, and opted for the metabolomics to inform switchgrass response to the drought. However, in our pilot experiments, we ensured timing and conditions such that both plants were pot-bound, i.e. covering the whole tube, prior to treatment initiation. This choice was made to ensure that root volume was as comparable as possible between the two plant species.

Major points

R1.3 Did I miss the data on the root biomass weights? How much of the differences observed between the two crops is explained by root biomass?

>>> Thank you for raising this important point regarding root biomass. As mentioned in the comment above (response R1.2), we could not collect root biomass data as the switchgrass roots were processed for metabolomics and we would not have data to compare to the bean root biomass. But referencing what we mention in response to comment R1.2., we believe that differences in root biomass will have minimal impact on the results of the study. We normalized root biomass by ensuring that the root system of the two crops was fully tube-bound before initiating treatments. Thus, even though there could be differences in root morphology and total root surface area, the two plants would roughly reflect similar root biomass and thus have minimal effects on the microbiome composition in the roots. However, we appreciate this point and have now added discussion of this caveat on PAGE 20.

R1.4 Switchgrass has 1/10 the shoot biomass of the beans and thus would be expected to have much less transportation and exudation.

>>> Thank you for bringing up this point. We agree that it is not necessarily straightforward that more shoot biomass is equivalent to more exudation, especially when comparing these two very different crops. In this study, we asked whether the effect of short-term drought may outweigh any crop-specific differences in microbiome responses, but this hypothesis was not supported and is one of the major take home results of the work, showing crop-specific microbiome responses in the face of short-term drought. Thus, any differences in transportation and exudation or any other plant traits (e.g., life history strategies, physiologies) likely collectively contributed to this outcome. We provide evidence that the switchgrass was changing its exudation profile during the drought as compared to the control as determined by the metabolomics analysis. We have now added mention of potential differences across the two plants in transportation and exudation (PAGE 20)

R1.5 Differences in crop biomass is likely also reflected in the soil moisture analyses which show that the soils in the tubes with the beans have much less soil moisture (the 'drought' is more severe).

>>> Thank you for this comment. We used the soil moisture data to confirm that the drought treatment was successful for both crops with drought soil having less moisture than the watered soil, but given that these are two different crops with different water needs, we purposely varied their watering by crop to provide a relatively equivalent stress to the plant, and this amount of water was informed by our pilot data. We expected that the extent of the soil moisture reduction during drought in bean would be greater than switchgrass because switchgrass is known for drought resistance. This means that the bean plants would take up more water from the soil to resist the drought. This was an expected result given we had specifically chosen plants that differ in their drought tolerance. This would also mean that the switchgrass crop/shoot biomass was expected to be less sensitive to drought treatment compared to bean plants, which was observed. We have clarified this information (PAGE 17).

R1.6 I find headings like “the most striking difference between the rhizobomes of the two crops was whether a plant was present” confusing because the rhizobiome is a rhizosphere community. I assume you are comparing the abundance of the microbes in the rhizosphere with their abundance in the soil and suggest that you clarify this because it's not clear after reading the methods.

>>> Thank you for this comment, we agree that this may be confusing. We have now clearly defined the rhizobiome as used in this study and clarify that differences the planted and unplanted conditions were all originated from root-influenced field soils, for each crop (page 5).

Again, the observation that there is a much smaller influence of the plant with SG could very well be due to the much lower root biomass.

>>> As mentioned in response to comments R1.2, R1.3 and R1.4, we did not have any observational evidence that the switchgrass had a lower root biomass, since roots of each plant were all tube-bound before the start of treatments and limited at that point by the size of the container. Also, in general switchgrass has a denser root system (a “rhizome”) than common bean. We have added text stating that we did not measure root biomass directly but observationally found them to be maximized to the size of the pot/tube (Page 20).

R1.7 Some discussion should be included on the potential bias introduced by using different soils to grow the two crops.

>>> Thank you for pointing out the difference, and we agree. We explain our reason to include different soils for the experiment on the second paragraph on page 17 (briefly, it would not be representative of or informative to the expected field conditions to use switchgrass soil with bean and vice versa given each soil’s crop-influenced microbiome legacy). We have added a few additional clarifications to make this clear.

R1.8 Make sure you have controlled for multiple comparisons. My understanding is that PERMANOVA does not control for false discovery when doing multiple comparisons and that it is recommended to use the Benjamini-Hochberg procedure.

>>> Thank you for this comment. As the reviewer suggests, a widely known fact in statistics is that the more tests are performed, the greater the chance of rejecting one or more null hypotheses simply by chance. For instance, if we were to perform 20 tests using an a priori significance level of 0.05, we would expect to reject a true null hypothesis, and get a P value smaller than 0.05 in one of those 20 tests by chance alone. However, the permutation *P*-values used in the PERMANOVA tests provide an exact test of each *individual* null hypothesis (see Anderson ¹ and Day and Quinn ²). Thus, *ad hoc* experiment-wise corrections (such as Bonferroni or Benjamini-Hochberg) would be inexact and overly conservative ². Therefore, we recognize this difference between multiple tests versus permutation tests and report the exact permutation *P*-values as statistically appropriate.

Minor points

R1.9 Line 48. A bit unclear what you are comparing here...how do you have a rhizobiome without a plant?

>>> Thank you, this is the same comment as R1.6. The originating soils were collected from plots actively growing the crops, and thus were root influenced, and so we refer to the microbiome of these soils as rhizobiomes. It was a key part of the design to use soils with a legacy of the crop so that the

responses of the crop-specific community could be assessed in the experiment. We have added a statement to the abstract pertaining to sampling root-influenced soil to make this clear in page 2 with a highlighted comment numbered R1.9.

R1.10 Line 136. More important vs. what? Dormant microbes wouldn't be expected to contribute much

>>> Thanks for this comment, we appreciate that a clarification is needed. The goal of the study was to understand active rhizobiome dynamics during short term drought. As mentioned in lines 151-153, we hypothesize that the role of the active community will be more important and prominent than the total or dormant pool in the immediate aftermath of stress. This will be a consequence of environmental fluctuations observed immediately after stress during which certain members of the rhizobiome can shift in competitiveness. As the reviewer rightly points out, dormant microbes will not be expected to contribute much during this time, hence we look at the active community in this study.

R1.11 Line 155. Is Red Hawk a drought sensitive line?

>>> Yes, Red Hawk is a drought sensitive line. We have added this with a reference (Dramadri Onziga et al. 2019 Crop Sciences) discussing drought on page 5.

R1.12 Suggest adding the ages of the plants to Figure 1. It's not obvious from the methods.

>>> Thank you for the suggestion, we have added the growth stages to the caption of Figure 1 (page 6) and emphasized them in the methods section.

R1.13 Was the brush used to remove the soil from the roots sterilized between plants or were new brushes used, should be clarified in the methods.

>>> Thank you, we used ethanol-cleaned gloves and brushed the soil from the roots using the sterilized gloves. Gloves were ethanol-cleaned again when handling new samples and new ethanol-cleaned gloves were used as needed (page 24).

R1.14 Line 573. Suggest adding a reference supporting using DNA concentration as a proxy for cell size.

>>> Thank you, included on page 10.

R1.15 The metabolomics analyses generally look good.

>>> Thank you for the positive feedback on the metabolomics section.

R1.16 The pooling of the metabolomics samples is confusing, why did you pool samples? Why include one replicate with 3 pooled samples? How did you account for this in the statistical tests?

>>> Thanks for pointing out that more clarification is needed here. Briefly, samples were pooled to balance sufficient plant tissue mass needed for metabolite extraction (0.1-0.5 g dry tissue) with retention of biological replicates needed for statistical tests. We have added more explanation (page 28) and refer also to **Supplementary Fig. 7**.

R1.17 Line 413. A bit confusing, I assume you scaled each metabolite feature by the maximum across all samples?

>>> Yes, the reviewer is correct. For each metabolite feature, we scaled its abundance by maximum (to a scale ranging from 0 to 1) across all samples. We have clarified this in the text (page 29).

R1.18 The metabolomics data must be deposited in a public repository. I suggest using GNPS because it will provide matches against real molecules and would be a nice complement to the CANOPUS and SIRIUS based annotations.

>>> The LC-MS metabolomics datasets are now publicly available on Dryad - 'Metabolomics data for Disentangling plant- and environment-mediated drivers of active rhizosphere bacterial community dynamics during short-term drought'. It has been assigned a digital object identifier (DOI): doi:10.5061/dryad.6t1g1jx5z. Prior to publication, you may get access to the data files using this URL: https://datadryad.org/stash/share/vkqIFDwKkWd0_U36mKWee1z1-pb3Xha3ajrZ7x_HO2c. We have updated the Data Availability statement accordingly on page 32.

We chose not to use GNPS because according to its policy, it only accepts data dependent acquisition (DDA) data.

<https://ccms-ucsd.github.io/GNPSDocumentation/isgnpsright/>

Although we could have deposited our DDA data alone to GNPS, in our case they were only used as an annotation file to support the DIA (data independent acquisition) datasets containing the feature abundance info obtained under Waters Q-ToF MS^E mode. The DDA data do not contain any metabolite abundance information. Therefore, it makes more sense to make the the DDA and DIA datasets available together as we have done.

R1.19 I'm only seeing feature annotations for the 50 features from the VIP analysis. Where are the CANOPUS/SIRIUS IDs for the rest of the features?

>>> It might be difficult to annotate all the features using CANOPUS/SIRIUS for this dataset. This is because our metabolomics data were collected from a Waters Q-ToF instrument and in turn processed using Waters Progenesis Q1 (Waters commercial software to process the untargeted metabolomics data). If the data were collected from a Thermo Orbitrap and processed by Compound Discoverer (Thermo's metabolomics software) or an open-source software (e.g., MZmine3), annotating the entire feature list with CANOPUS/SIRIUS IDs would be feasible given the software compatibility.

In our case, we separately imported our DDA data (with the MS/MS fragmentation info) to a standalone version of CANOPUS/SIRIUS to get the annotations. We, then, manually match them back to the features on the list generated by the Progenessi Q1 (the same approach reported in our previous publication, <https://doi.org/10.1021/acs.jafc.2c01306>). Doing this for the entire feature list will be time-consuming (and perhaps not necessary), as we focus on the features changing significantly (according to the VIP analysis) across the biological samples in the current study. For those features that were not changing significantly, we did not perform the manual annotation.

R1.20 Double check that the 50 features from the VIP analysis are indeed unique features. For example, 10.52_678.3852n and 10.76_678.3851n seem like they could be resulting from the feature finding algorithm. Are they baseline resolved?

>>> We predict these are two individual features (rather than the result of an algorithm issue), although their calculated neutral mass appears to be almost identical. The identical neutral mass and close retention time are usually seen for structural or stereoisomers. To the best of our knowledge, this is not a rare case for metabolomics analysis of plant tissue extracts.

Our prediction was also based on the following observation. **Fig.1 below shows an example of two different ion species generated from the same metabolite.** Feature 2 (with a miscalculated neutral mass by software) is in fact a dimer of feature 1. As you can see the abundances of these two features across the different samples are almost perfectly correlated to each other. If we predict the 10.52_678.3852n and 10.76_678.3851n are the same feature resulting from the feature finding algorithm, we expect to see a similar (or even more perfect) linear correlation between them. However, **Fig. 2** shows that there are non-linear discrepancies in the abundances of these two features across the samples.

Fig. 1 Correlation of 11.72_300.2099 (feature 1) and 11.72_292.2047n (feature 2) based on their abundances across the samples.

Fig. 2 Correlation of 10.52_678.3852n (feature 1) and 10.76_678.3851n (feature 2) based on their abundances across the samples.

R1.21 When you sort the 50 by retention time you see that many of them are very close in RT.

>>> Thanks for this comment. The features with close RT are commonly seen for the plant metabolites detected by an untargeted metabolomics using plant tissues extracts as the input samples. This observation indicates that many of the identified metabolites in the switchgrass rhizosphere are similar

in polarity (hydrophobicity) and might also be structurally related and derived from the same biosynthetic precursors (or pathways).

R1.21 The MS/MS for the 10.11_1000.5271n is very messy and does not provide strong evidence, the text should mention that the MS/MS spectra is low intensity but suggests that this may be the same ion detected in the previous study.

>>> We thank the reviewer for this comment. We have revised the text to draw the reader's attention. Please see page 9.

R1.22 Note that S14 from that previous study annotates the feature as “1 x hexose, 2 x deoxyhexose, 1 x pentose” so I think the name mono-glycosylated is a bit confusing (though this is likely used to indicate that it is glycosylated at one position). Suggest ‘glycosylated steroidal saponin’.

>>> Yes, we thank the reviewer for this comment. Calling this saponin a “mono-glycosylated” saponin can be misleading. We have now used the term “the saponins with a single sugar-chain” instead to accurately describe this group of saponins.

Reviewer #2 (Remarks to the Author):

R2.1 Bandopadhyay et al investigated bacterial community and plant root metabolites in soils planted or unplanted by bean or switchgrass during a course of short-term drought. Effect of plant presence, plant identity, and drought on bacterial community is detected. Effect of drought on metabolite is also detected. Overall, this reviewer thinks the manuscript is not ready for publication in Nat Comm. Most of the analysis are descriptive and superficial, and the rationale for the experiment design is not well-documented.

>>> Thanks for your time to review the work and for your thoughtful feedback. We note that this comment contrasts with the other reviewers, who commented that the manuscript is well written with clear figures and text and interesting results (R1) as well as a well-designed and written study (R3). Regardless, we appreciate the encouragement to strengthen the clarity and writing and have revised throughout to do so, so that the rationale, design and analysis of this study are clear.

R2.2 The results of metabolites and bacterial community remain independent from each other.

>>> Thank you for this comment, and we agree. The reason that the metabolite and bacterial community data are independent are because of the biological outcomes: there were statistically

supported changes in the metabolite profile of switchgrass but no changes in the rhizobiome. Thus, we could not use one to inform the other because the results suggest that they are responding independently for the switchgrass. The discordance between the metabolites and the bacterial responses is one of the main results of the study. This is elaborated in page 18 comment bubble R2.2.

R2.3 Fig. 1 The experiment design is not clearly described in the figure legend. Why metabolomics sampled for switchgrass, but viability assays for bean?

>>> Thank you for this comment. We have revised the figure legend to describe these better (page 6). Please also refer to pages 8 and 24 (comment bubble R2.3) for more details on the experimental design.

R2.4 Fig. 3 What is VIP. What's PLS-DA? Figure legend should be self explainable.

>>> Thank you, we have updated the Fig 3 legend (page 10) to clarify: VIP: Variable of importance, PLS-DA: Partial Least-Squares Discriminant Analysis.

R2.5 Line 48-50. However, the presence of a plant more strongly explained the rhizobiome variation in bean (17%) than in switchgrass (3%), with a small effect of plant mediation during drought only observed for the bean rhizobiome. May the difference of bean and switchgrass due to the difference in the moisture between the bean and switchgrass treatments. As the figure 2 showed that the measured soil moisture varied between bean and switchgrass.

>>> Thank you for this comment. The difference in soil moisture across plants was a feature of the experimental design because of the different drought sensitivities of each plant type. Prior to running the full experiment, we conducted a pilot test of each plant type to determine the progressive water levels needed to stress them without killing them. We found that, as expected, the water requirement was different between bean and switchgrass plants. We knew that the bean plants would take up more water from the soil to resist the drought and cause a greater decline in soil moisture compared to switchgrass. This was an expected result given we had specifically chosen plants that contrast in their drought tolerance. Please also see response to R1.5 and the added text on page 17.

R2.6 Line 45-46. there were differences in the active microbiome structure between drought and watered and between planted and unplanted treatments. This is already known by a lot of previous studies, and this study provided no novelty.

>>> Thank you for this comment. We are not aware of any other studies that have directly compared drought/watered and planted/unplanted in a factorial design, and across two different crop plants manipulated comparably, and assessing specifically the active microbiome (rather than total DNA, which includes active, inactive, and dead). We are also not aware of a study that showed the independence of metabolite and rhizobiome drought response. Please see page 18.

R2.7 Despite their different community structures, the drought rhizobiome dynamics were similar across the two crops. Not clear about how which results support this conclusion?

>>> Thank you for this comment. The results supporting this conclusion is provided in Supplementary Fig.5 which show linear models of the active rhizobiome structure (beta diversity) over time and as compared to the pre-drought condition. These results are elaborated in page 12 and are highlighted in a comment bubble numbered R2.7.

R2.8 Line 54 its risky to conclude 'decoupling of plant exudation and rhizobiome responses'.

>>> Thank you, we agree that this observation from our results merits further investigation, and deliberately clarified/tempered the statement with "possibility" of decoupling in the abstract (page 2). Our results indicate that the switchgrass rhizobiome and metabolite profiles show different responses to drought treatments leading us to this conclusion.

R2.9 Fig. 4 Fig. 1 is hard to read. This reviewer suggest not to use different size to depict harvest time.

>>> Thank you for the comment, we're sorry that you find the figure difficult to read and we agree that it is intentionally rich in information so it can convey all aspects of the experimental design without overcrowding the individual experimental points. For Figure 4, symbol size was chosen to represent time because it is a continuous variable. Because there are limited options for distinguishing continuous variables, we opted to keep this choice as is. If the reviewer has a specific suggestion to replace the size with a different way of distinguishing the continuous variable, we are very open and happy to receive the idea.

R2.10 Why the samples of beans formed two cluster?

>>>Thanks for the question. As discussed on page 13, the two clusters are indicative of the strong effect of planted versus unplanted treatments in bean plants. This effect was not as strong for switchgrass.

R2.11 Fig. 5-6 Why bean only? No switchgrass?

>>> As specified on page 13, we only show bean plants in figures 5-6 because there was no statistically-supported plant-mediated effect of drought observed in the switchgrass.

R2.12 Fig. S7, low resolution, hard to read, what's the different bars at the horizontal axis?

>>> Thank you for this comment, we have updated to provide a higher-resolution copy of Supplementary Fig. 4 (previously S7). To clarify, the bars on the horizontal axes show the bacterial community composition at "Class" level, as described in the figure caption.

Reviewer #3 (Remarks to the Author):

General Comments:

R3.1 In this study, the authors conducted greenhouse studies of common bean and switchgrass to evaluate the bacterial rhizobiome responses to short-term drought (6 day) using 16S rRNA and 16S rRNA gene sequencing to determine if 1) active rhizobiomes respond to drought and change progressively with drought severity and/or 2) the host plant mediates the responses of the active rhizobiome to drought via compositional changes in root metabolites. In addition, metabolite data was collected for switchgrass and plant performance for bean (switchgrass did not demonstrate differences for plant performance for the short-term drought plants compared to watered). Rhizobiome community analysis for the 'active' fraction of the community (OTUs for which their 16SrRNA:rRNA gene ≥ 1) showed that there were statistically supported differences in community structure with drought and time/drought severity for both switchgrass and bean, but that statistical difference in temporal dynamics was not observed. Therefore, the authors determined that hypothesis 1 had been supported by their data. With regard to hypothesis 2, the authors found that both plants influence the rhizobiome, but the bean rhizobiome showed greater response to the presence of the plant (17% explanatory value) compared to switchgrass (3% explanatory value). However, they found that only the bean rhizobiome had a significant structure change in response to drought. Thus, the authors indicated that hypothesis 2 was only partially supported by their data. Because the bean rhizosphere did appear to change in response to drought, the most abundant activated taxa that were uniquely associated with the planted drought condition in bean rhizobiomes were also identified (87 taxa total). Overall this is a well-designed and written study that describes how the switchgrass and bean rhizobiomes respond to a short duration drought under greenhouse-grown conditions. However, the manuscript can be improved with attention to the following items described below.

>>> Thank you for your time in providing the review and for your positive comments. We are glad that you find this to be a well-designed and well-written study, and thank you for the suggestions for improvement.

Specific Comments for the Authors' Consideration:

R3.2 The authors designed the plant growth set-up such that the plants were root-bound by the start of applying drought conditions so that it is assumed that all the soil in the pots is in close proximity to soil and can be considered rhizosphere soil. While this setup does simplify root sampling, it is known that root architecture and spatial arrangement does impact rhizobione recruitment and maintenance, so the concern then is, how well does the data generated in this greenhouse study reflect rhizobiome dynamics that would occur in a 6 day-drought scenario in the field for switchgrass and common bean? The authors should address this concern in their discussion.

>>> Thank you for pointing out this important factor, and we agree that the results from lab-based experiments may not always reflect field conditions, but are critical to direct and formulate hypotheses which can be then tested in field studies. We have now included a discussion pertaining to this aspect in the manuscript page 21.

R3.3 The authors explain that they used a 6-day drought because they were interested in understanding how rhizobiome communities shift under a short duration drought scenario for a relatively drought resilient (switchgrass) and more drought sensitive plant (common bean) to better understand initial and short-term community changes. The authors should include information from other studies (if available) about how switchgrass and common bean have been shown to respond to short versus longer term drought to provide further rationale for why the 6 day drought period was used.

>>> Thank you for this question. We elaborate on our rationale of selecting the 6-day drought duration on page 5. Briefly, we were interested in the immediate activity responses (activation or deactivation) of rhizobiome members to the stress and in teasing apart their responses to drought from that of their responses to the host. Based on expectations of soil bacterial replication and community turnover (replacement of members), our 6-day study should capture only immediate activity switches of existing populations and at most one generation of any cells that are selected by the experimental conditions. In contrast, longer term studies that have been done have focused on the outcomes of community assembly (selection including by environmental fluctuations into a recovery period, competition, immigration, extinction).

We searched the literature to find studies that specifically compared short and long-term drought exposure for switchgrass or bean and could not find relevant studies. We did find several studies using long-term drought exposure beyond 6 days, which are included in the references of the abovementioned section on page 5.

R3.4 The authors do include commentary on why they relied on 16S rRNA and 16S rRNA gene sequencing to examine rhizobiome changes in response to drought and what the limitations of that methodology can be. I think the authors did develop solid, interpretable data using this method; however, had the authors used a genome-resolved metagenomics analysis approach, then it would be likely that rhizobiome metabolisms that play a role in a productive drought response could be identified, and this would have greater impact to the field.

>>> Thank you for the positive comments on the data and methodology presented in this paper. We are glad that you found the data to be solid and interpretable. We agree that a genome resolved metagenomics approach could help in the identification of metabolic pathways that play a role in a productive drought response and thus could have greater impact to the field. Based on our experience in using genome-resolved metagenomics (for example, please see Howe et al. 2023 Nature Communications), we did not want to apply this method *initially* for two reasons. The first reason is that we were not sure if there would be a strong plant-mediated drought response at the short-term exposure (it was a hypothesis that we tested in this study), so we wanted to understand if there were

observable responses before investing in the substantially more resources needed to understand any metabolic pathways involved (which we now see as a next step for the bean, but not the switchgrass).

The second reason has to do with some of the known technical challenges of MAGs-based transcriptomics. For example, the recruitment of transcripts of functional genes to MAGs is heavily biased towards the most abundant members, which are more likely to have high-quality/low contamination MAGs returned, and these abundant members may not necessarily be the active members in drought conditions. Based on our experience, it is difficult to get enough mRNA from the droughted soil extracts to have confidence in their observation (coverage and mapping to the MAGs). There would have been highly uneven confidence/coverage of the mRNA from the watered soils as compared to the droughted, making standardization for cross-comparisons potentially difficult. Similarly, there is a technical challenge with host read contamination interfering with the mRNA signal. In summary, even with our extensive experience in applying genome-resolved metagenomics in other challenging systems, we had remaining concerns about several technical issues specific to the drought condition of the experimental design that may ultimately render the data uninterpretable. We agree that it would be a great next step, but we would have to critically examine some of these technical issues and resolve them confidently before moving forward.

R3.5 It is not clear from the study design and discussion why the examination of the fungal rhizobiome community was not also included in the analysis since it is known that fungi (especially AM fungi) contribute to the drought response in switchgrass and bean. The authors should also address this limitation in their study discussion.

>>> Thank you for this key comment, and we agree that AM fungi are shown in the literature to be important in drought tolerance. We now include a discussion on this limitation of the study in the discussion section in page 21.

References

- 1 Anderson, M. PERMANOVA+ for PRIMER: guide to software and statistical methods. *Primer-E Limited*. (2008).
- 2 Day, R. & Quinn, G. Comparisons of treatments after an analysis of variance in ecology. *Ecological monographs* **59**, 433-463 (1989).

REVIEWERS' COMMENTS

Reviewer #1 (Remarks to the Author):

In general, the authors addressed my concerns (though it took some hunting to find all of the revisions). However, there are a few points below that need a bit more attention (see my comments in ALL CAPS below).

We chose not to use GNPS because according to its policy, it only accepts data dependent acquisition (DDA) data.

<https://ccms-ucsd.github.io/GNPSDocumentation/isgnpsright/>

Although we could have deposited our DDA data alone to GNPS, in our case they were only used as an annotation file to support the DIA (data independent acquisition) datasets containing the feature abundance info obtained under Waters Q-ToF MSE mode. The DDA data do not contain any metabolite abundance information. Therefore, it makes more sense to make the the DDA and DIA datasets available together as we have done.

REGARDING “THE DDA DATA DO NOT CONTAIN ANY METABOLITE ABUNDANCE INFORMATION. “
I’M UNLCLEAR HOW YOU DID DDA ANALYSIS W/O COLLECTING MS1 SCANS. PLEASE DESCRIBE THE DIA METHOD YOU USED IN THE METHODS SECTION AND THE POTENTIAL LIMITATIONS OF ANALYZING THE DATA IN THIS WAY.

R1.20 Double check that the 50 features from the VIP analysis are indeed unique features. For example, 10.52_678.3852n and 10.76_678.3851n seem like they could be resulting from the feature finding algorithm. Are they baseline resolved?

>>> We predict these are two individual features (rather than the result of an algorithm issue), although their calculated neutral mass appears to be almost identical. The identical neutral mass and close retention time are usually seen for structural or stereoisomers. To the best of our knowledge, this is not a rare case for metabolomics analysis of plant tissue extracts.

Our prediction was also based on the following observation. Fig.1 below shows an example of two different ion species generated from the same metabolite. Feature 2 (with a miscalculated neutral mass by software) is in fact a dimmer of feature 1. As you can see the abundances of these two features across the different samples are almost perfectly correlated to each other. If we predict the 10.52_678.3852n and 10.76_678.3851n are the same feature resulting from the feature finding algorithm, we expect to see a similar (or even more perfect) linear correlation between them.

However, Fig. 2 shows that there are non-linear discrepancies in the abundances of these two features across the samples.

PLEASE CONFIRM THAT YOU CHECKED ALL OF THE OTHER FEATURES TO MAKE SURE THEY ARE NOT REDUNDANT

R1.21 The MS/MS for the 10.11_1000.5271n is very messy and does not provide strong evidence, the text should mention that the MS/MS spectra is low intensity but suggests that this may be the same ion detected in the previous study.

>>> We thank the reviewer for this comment. We have revised the text to draw the reader's attention. Please see page 9.

REVISE TEXT TO STATE THAT THE MS/MS IS NOT CLEAR BUT YOU SPECULATE THAT IT'S THE SAME COMPOUND. SUGGEST INCLUDING THE MS/MS SPECTRA FROM THE REF 8 IN SI FIGURE 2

Reviewer #2 (Remarks to the Author):

The authors addressed all the concerns of my last review.

Reviewer #3 (Remarks to the Author):

In their revised submission, the authors adequately addressed reviewer concerns. The revised figure legend for the experimental design depicted in Figure 1 does better explain the experiment design and the rationale for the design, and updates to Figure 2 now provide soil moisture and shoot biomass for each for samples for each treatment-time combination. The authors also now include discussion of the limitations of the study with regards to accounting for the impacts of root architecture and spatial arrangement and the involvement of fungal rhizobiome members on the drought response.

Key:

- Yellow highlighted text: reviewer's comment from the last review.
- **Yellow highlighted/bold text: reviewer's comment from the current review**
- Blue text: my response for the last review
- Red text: my response for the current review

REVIEWERS' COMMENTS

Reviewer #1 (Remarks to the Author):

In general, the authors addressed my concerns (though it took some hunting to find all of the revisions). However, there are a few points below that need a bit more attention (see my comments in ALL CAPS below).

We chose not to use GNPS because according to its policy, it only accepts data dependent acquisition (DDA) data (<https://ccms-ucsd.github.io/GNPSDocumentation/isgnpsright/>). Although we could have deposited our DDA data alone to GNPS, in our case they were only used as an annotation file to support the DIA (data independent acquisition) datasets containing the feature abundance info obtained under Waters Q-ToF MSE mode. The DDA data do not contain any metabolite abundance information. Therefore, it makes more sense to make the DDA and DIA datasets available together as we have done.

REGARDING “THE DDA DATA DO NOT CONTAIN ANY METABOLITE ABUNDANCE INFORMATION. “ I’M UNLCLEAR HOW YOU DID DDA ANALYSIS W/O COLLECTING MS1 SCANS. PLEASE DESCRIBE THE DIA METHOD YOU USED IN THE METHODS SECTION AND THE POTENTIAL LIMITATIONS OF ANALYZING THE DATA IN THIS WAY.

Sorry for the confusion with my original response. The DDA method certainly does acquire MS1 survey scans before picking the top 5 ions for MS/MS. However, we only analyzed a pooled sample with the DDA method. All other individual samples were analyzed using the DIA method. The metabolite abundance information comes from DIA data, and the DDA data from the pooled sample is only used to support compound annotations.

R1.20 Double check that the 50 features from the VIP analysis are indeed unique features. For example, 10.52_678.3852n and 10.76_678.3851n seem like they could be resulting from the feature finding algorithm. Are they baseline resolved?

We predict these are two individual features (rather than the result of an algorithm issue), although their calculated neutral mass appears to be almost identical. The identical neutral mass and close retention time are usually seen for structural or stereoisomers. To the best of our knowledge, this is not a rare case for metabolomics analysis of plant tissue extracts.

Our prediction was also based on the following observation. Fig.1 below shows an example of two different ion species generated from the same metabolite. Feature 2 (with a miscalculated neutral mass by software) is in fact a dimer of feature 1. As you can see the abundances of these two features across the different samples are almost perfectly correlated to each other. If we predict the 10.52_678.3852n and 10.76_678.3851n are the same feature resulting from the feature finding algorithm, we expect to see a similar (or even more perfect) linear correlation between them. However, Fig. 2 shows that there are non-linear discrepancies in the abundances of these two features across the samples.

PLEASE CONFIRM THAT YOU CHECKED ALL OF THE OTHER FEATURES TO MAKE SURE THEY ARE NOT REDUNDANT

We confirm that all the top 50 PLS-DA features are unique features.

R1.21 The MS/MS for the 10.11_1000.5271n is very messy and does not provide strong evidence, the text should mention that the MS/MS spectra is low intensity but suggests that this may be the same ion detected in the previous study.

We thank the reviewer for this comment. We have revised the text to draw the reader's attention. Please see page 9.

REVISE TEXT TO STATE THAT THE MS/MS IS NOT CLEAR BUT YOU SPECULATE THAT IT'S THE SAME COMPOUND. SUGGEST INCLUDING THE MS/MS SPECTRA FROM THE REF 8 IN SI FIGURE 2

We thank the reviewer for the suggestion and have revised the text accordingly. We also included the MS/MS spectra of same saponin feature detected in switchgrass roots that we speculate the 10.11_1000.5271n to be in the new Supplemental Figure 2.